# A gene cluster in *Ginkgo biloba* encodes unique multifunctional cytochrome P450s that initiate ginkgolide biosynthesis

Victor Forman [1], Dan Luo[1], Fernando Geu-Flores [1], René Lemcke [2], David R. Nelson[3], Sotirios C. Kampranis [1], Dan Staerk [4], Birger Lindberg Møller [1] & Irini Pateraki [1] ✉

The ginkgo tree (*Ginkgo biloba*) is considered a living fossil due to its 200 million year's history under morphological stasis. Its resilience is partly attributed to its unique set of specialized metabolites, in particular, ginkgolides and bilobalide, which are chemically complex terpene trilactones. Here, we use a gene cluster-guided mining approach in combination with co-expression analysis to reveal the primary steps in ginkgolide biosynthesis. We show that five multifunctional cytochrome P450s with atypical catalytic activities generate the *tert*-butyl group and one of the lactone rings, characteristic of all *G. biloba* trilactone terpenoids. The reactions include scarless C–C bond cleavage as well as carbon skeleton rearrangement (NIH shift) occurring on a previously unsuspected intermediate. The cytochrome P450s belong to CYP families that diversifies in pre-seed plants and gymnosperms, but are not preserved in angiosperms. Our work uncovers the early ginkgolide pathway and offers a glance into the biosynthesis of terpenoids of the Mesozoic Era.

The ginkgo tree (*Ginkgo biloba*, ginkgo, maidenhair tree), with its characteristic fan-shaped and leathery leaves, has served as cultural inspiration throughout human history[1]. Fossil records dating back over 200 million years show that this iconic tree is the single living member of the once large Ginkgoaceae family, most of which did not survive the Pleistocene glaciations[1,2]. Throughout this extended period, the leaf morphology of ginkgo remained unchanged (morphological stasis), thus earning the classification as 'living fossil'[1,3,4]. Stasis in morphology does not necessarily imply stasis in biochemistry; nevertheless, it is evident that ginkgo trees produce unique bioactive specialized metabolites not encountered in modern plants. Characteristic examples are the ginkgolides and bilobalide (Fig. 1a), which are terpene trilactones that may have contributed to the resilience and survival of this distinct plant species[5]. In particular, ginkgolides display complex chemical structures, including six 5-membered rings, three lactone rings and a *tert*-butyl group[6,7]. The pharmaceutical and nutraceutical properties of

ginkgolides and bilobalide are numerous and are associated largely with their ability to penetrate the blood–brain barrier[6,8]. These properties include neuromodulatory effects, increased cerebral blood flow and circulation, modification of neurotransmission, protection against neural cell apoptosis, and anti-inflammatory activities, partly due to platelet-activating factor (PAF) and $GABA_A$ receptor antagonistic activity[9–12]. Standardized *G. biloba* extracts (EGb761) are one of the best-selling food supplements today, with a global market value expected to reach 15.26 billion USD by 2028[13].

Although ginkgo terpenoids have been studied extensively for their pharmaceutical properties, knowledge of their biosynthesis remains limited. The only enzyme proposed to be involved in the biosynthesis of ginkgolides is *Gb*LPS (levopimaradiene synthase), a diterpene synthase that catalyzes the synthesis of levopimaradiene (**2**) —the proposed precursor of all ginkgolides[14] (Fig. 1b). It has been suggested that cytochrome P450 monooxygenases (CYPs) also

[1]Faculty of Science, Department of Plant and Environmental Sciences, Section for Plant Biochemistry, University of Copenhagen, Copenhagen, Denmark. [2]Faculty of Health and Medical Sciences, Department of Neuroscience, University of Copenhagen, Copenhagen, Denmark. [3]Faculty of Microbiology, Immunology and Biochemistry, The University of Tennessee Health Science Center, University of Tennessee, Memphis, TN, USA. [4]Faculty of Health and Medical Sciences, Department of Drug Design and Pharmacology, University of Copenhagen, Copenhagen, Denmark. ✉e-mail: eipa@plen.ku.dk

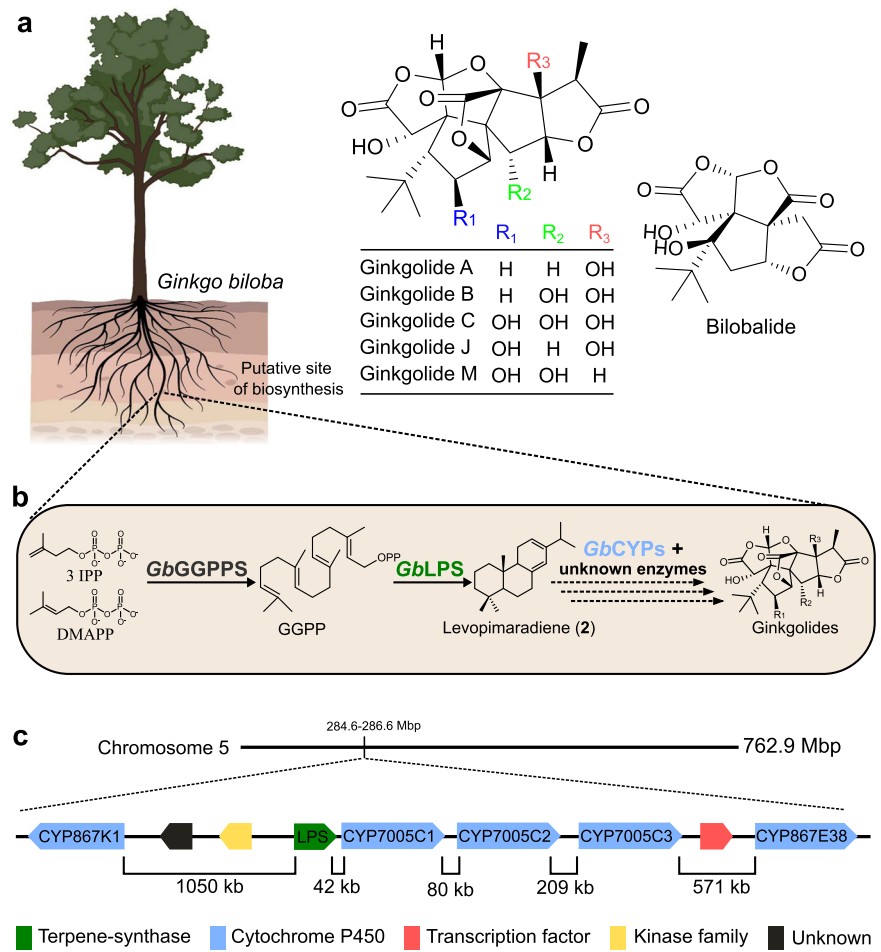

**Fig. 1 | Biosynthesis of ginkgolides. a** Chemical structure of ginkgolides (diterpenoids) and bilobalide (sesquiterpenoid). The tree image has been created by BioRender.com (2021). **b** Ginkgolide biosynthesis is suggested to take place in the roots of the tree, as indicated by the dashed lines. Geranylgeranyl pyrophosphate synthase (*Gb*GGPPS) is producing geranylgeranyl pyrophosphate (GGPP), which initiates the biosynthesis of all diterpenoids. Next, levopimaradiene synthase (*Gb*LPS) converts GGPP to diterpene hydrocarbon levopimaradiene, which is the hypothesized precursor of ginkgolides. It has been proposed that levopimaradiene is oxygenated by cytochrome P450s (CYPs) en route to ginkgolides. Further rearrangements in the carbon skeleton may be catalyzed by CYPs in connection with oxidation or by yet unknown enzymes. **c** Organization of the identified biosynthetic gene cluster (BGC) on chromosome 5. The *GbLPS* gene is located in close proximity to five *CYP* genes. Genes coding for a putative kinase, a GTE10-like transcription factor, and an unknown protein are also present (indicated by different colored boxes). Mbp million base pairs, kb kilo base pairs.

participate in the biosynthesis of these compounds, as inhibition of CYPs in ginkgo seedlings was shown to decrease the accumulation of ginkgolides[15]. Ginkgo terpenoids accumulate in the entire plant, however, their biosynthesis has been hypothesized to take place in the roots, where *GbLPS* is mainly expressed[14] (Fig. 1a, b). Despite numerous metabolomics studies conducted on ginkgo extracts[16], to the best of our knowledge, no pathway intermediates have been identified for ginkgolides, suggesting that the pathway is highly channeled[17]. This, in combination with their unique chemical structures (Fig. 1a), has made it challenging to infer possible biosynthetic routes.

In this work, we unveil the first steps in the biosynthesis of ginkgolides following a combination of biosynthetic gene cluster (BGC) mining and gene co-expression analysis. We choose these strategies given that genes coding for enzymes in specialized metabolite pathways often tend to be co-expressed and are sometimes arranged in BGCs in the genome[18–20]. Upon mining the publicly available *G. biloba* genome[21], we identify five cytochrome P450 (CYP)-encoding genes in close proximity to the *GbLPS* gene. Functional characterization of these CYPs shows multifunctional enzymes with unprecedented activities, including scarless C−C bond cleavage and a carbon skeleton rearrangement (NIH shift) occurring on a previously unsuspected pathway intermediate. Combining the BGC findings with

transcriptomic data, we show that the CYPs located in this BGC have similar expression patterns, not only to *GbLPS*, but also to an additional CYP encoding gene that likely forms the first lactone ring of ginkgolides. Of the five CYPs encoded in the revealed putative BGC, three belong to the CYP7005 family, which has only been identified in pre-seed plants (ferns)[22]. The other two belong to the CYP867 family, which is exclusive to gymnosperms and whose members have not been functionally characterized yet[23]. Our work demonstrates that co-expression analysis and the mining of BGCs are valuable tools for pathway elucidation, even in early diverging plants with large genomes like the ginkgo tree. This work establishes the early steps in the biosynthetic pathway towards ginkgolides, offers a glance into the biosynthesis of terpenoids of the Mesozoic era, and supports the characterization of *G. biloba* as a 'living fossil'.

## Results
### The *GbLPS* and five *GbCYP* genes form a biosynthetic gene cluster

To identify candidate genes potentially involved in ginkgolide biosynthesis, we searched for the *GbLPS* in the publicly available *G. biloba* genome draft[21] and mined the surrounding genomic region for genes encoding putative biosynthetic enzymes. We found *GbLPS* in

chromosome 5, in close proximity to five CYP encoding genes: *GbCYP7005C1*, *GbCYP7005C2*, *GbCYP7005C3*, *GbCYP867K1* and *GbCYP867E38* (Fig. 1c). *GbCYP7005C1*, *GbCYP7005C2* and *GbCYP7005C3* are positioned in tandem, right next to *GbLPS*. *Gb*CYP7005C1 and *Gb*CYP7005C3 share >98% amino acid sequence identity, while *Gb*CYP7005C2 shares ~90% sequence identity with *Gb*CYP7005C1 and *Gb*CYP7005C3. *Gb*CYP867K1 and *Gb*CYP867E38 share 45% sequence identity. In the vicinity, we also identified genes coding for a putative kinase and a putative GTE10-like transcription factor (TF) (Fig. 1c), as well as a gene of unknown function and a number of pseudogenes and retrotransposon elements (mainly Ty1-copia-like and Ty3-Gypsy-like LTRs).

The genomic association of the *GbLPS* gene with five CYP-encoding genes implies the presence of a BGC relevant to the biosynthesis of ginkgolides, since CYP enzymes are typically involved in the formation of oxygenated terpenoids[24], and inhibition of CYPs in ginkgo seedlings has been shown to decrease the accumulation of ginkgolides[15]. Ginkgolides are likely defense-related compounds in *G. biloba* and the TF present in the putative BGC may serve as a transcriptional regulator of the genes involved in their biosynthesis, since a homolog in Arabidopsis[25] has been shown to regulate responses to stress and environmental changes. Similarly, the kinase-like encoded enzyme could potentially regulate ginkgolide biosynthesis through phosphorylation.

## Unprecedented CYP activity drives the formation of the *tert*-butyl moiety present in all ginkgolides

To determine whether the CYPs encoded in the putative BGC participate in ginkgolide biosynthesis, we cloned the corresponding coding sequences from root cDNA and tested them in combination with *Gb*LPS using two different but complementary expression systems: *Nicotiana benthamiana* (tobacco)[26,27] and *Saccharomyces cerevisiae* (yeast)[28] (Fig. 2). *Gb*LPS is a multifunctional diterpene synthase that, independently of the expression host (Fig. 2a, c), produces a mixture of up to six diterpene hydrocarbons (compounds **1**–**6** in Fig. 2b) with levopimaradiene being the main product[27,29] (Fig. 2a–c; Supplementary Fig. 1).

We first expressed each individual CYP together with *Gb*LPS transiently in tobacco (Supplementary Method 2) and analyzed the accumulated terpenoids by liquid chromatography coupled with high-resolution mass spectrometry (LC–HRMS). Both *Gb*CYP7005C1 and *Gb*CYP7005C3 led to the accumulation of the same compounds, **7** ($C_{20}H_{30}O_2$) and **8** ($C_{20}H_{30}O_1$), with molecular formulas indicating oxidative modifications of a *Gb*LPS product (Fig. 2d; Table 1 and Supplementary Figs. 2, 3). Thus, the high sequence identity of these two CYPs was reflected in identical catalytic activities. For the remaining three CYPs of the identified BGC, only *Gb*CYP7005C2 produced minor amounts of **8** when co-expressed with *Gb*LPS (Supplementary Fig. 4c, d; Supplementary Data 1).

Next, we stably expressed the same combinations of enzymes in yeast via genomic integration. To enable efficient and stable expression of multiple heterologous genes in this host, new vectors were constructed for genomic integration based on previously reported plasmids[30,31] that utilize the defined and well-established chromosomal loci X-2, X-3, X-4, XI-2, XI-5, XII-2 and XII-5 (Supplementary Method 3). To generate yeast strains producing high amounts of diterpenoids, we co-expressed GGPP synthesis-boosting genes (e.g. the GGPP synthase *Sp*GGPPS7, and the truncated 3-hydroxy-3-methylglutaryl-coenzyme A reductase from yeast, *Sct*HMGR)[32] with variants of codon-optimized *Gb*LPS (Supplementary Method 4). This afforded diterpenoid production up to 146 mg/L (Fig. 2e; Supplementary Table 1). To support the activity of the heterologously expressed *Gb*CYPs in yeast, we co-expressed two cytochrome P450 reductases (PORs) identified in *G. biloba* transcriptomes, *Gb*POR1 and *Gb*POR2. The two *Gb*PORs were separately introduced into the optimized yeast strain expressing

*Gb*LPS (sGIN4) together with *Gb*CYP7005C1 and *Gb*CYP7005C3 (Supplementary Table 2). As in tobacco, LC–HRMS analysis identified **7** as the main product of these strains, together with **8** (Fig. 2f) and a small amount of a potential hydrogenated form of **8**, compound **9** ($C_{20}H_{32}O_1$). Strains expressing *Gb*POR2 showed increased production of **7** by more than 200% compared with the strains harboring *Gb*POR1 (Supplementary Fig. 5a). Accordingly, *Gb*POR2 was selected for inclusion in subsequent studies.

From an up-scaled culture of the yeast strain expressing *Gb*CYP7005C1 and *Gb*POR2 (sGIN8), compounds **7**–**9** were isolated and structurally characterized by NMR spectroscopy (Supplementary Figs. 16–30; Supplementary Tables 3–5). Compound **7** was identified as a *Gb*LPS product in which the decalin ring had been opened at two different positions, one of the openings resulting in the formation of the *tert*-butyl group present in all ginkgo terpene lactones. We named this compound ginkgosinoic acid A (Fig. 2g). Compounds **8** and **9** corresponded to 2-hydroxy derivatives of **3** and **2**, respectively, suggesting that the substrate of *Gb*CYP7005C1 and *Gb*CYP7005C3 is either levopimaradiene (**2**) or dehydroabietadiene (**3**). To investigate this, we made use of a pair of diterpene synthases from *Coleus forskohlii*, *Cf*TPS1 and *Cf*TPS3, which are unable to produce **2** but produce **3**–**6** together with miltiradiene (**10**)[33] (Supplementary Fig. 5b–d). When replacing *Gb*LPS with *Cf*TPS1 and *Cf*TPS3, we did not identify **7**–**9** in the strains expressing *Gb*CYP7005C1 and *Gb*CYP7005C3. Thus, **2** appears to be the only substrate utilized by *Gb*CYP7005C1 and *Gb*CYP7005C3, and is likely, the precursor of all ginkgolides.

The chemical structures and apparent relative abundances of **7**–**9** suggest that **7** is the final product of *Gb*CYP7005C1 and *Gb*CYP7005C3, and that **8** and **9** are either intermediates or by-products. To produce additional insights into the reaction sequence of these enzymes, we fed **8** and **9** separately to yeast strains expressing each of the CYPs as well as *Gb*POR2 (but not *Gb*LPS). The results showed that only **9** can be used by *Gb*CYP7005C1 and *Gb*CYP7005C3 for the synthesis of **7**, and it can also oxidize to **8** either spontaneously or aided by endogenous yeast enzymes (Supplementary Fig. 5e). The conversion of **9** to **8** is an aromatization reaction similar to the spontaneous one observed by Zi and Peters for a similar diterpenoid[34]. All in all, our results suggest that *Gb*CYP7005C1 and *Gb*CYP7005C3 are redundant and multifunctional, each being able to convert **2** into **9**, and then **9** into **7** (see below-suggested reaction mechanism). The overall conversion of **2** to **7** is a chemical transformation that involves double C–C cleavage of decalin system (at C2–C3 and at C9–C10) leaving no evident scar on the *tert*-butyl group. To explain this reaction, we propose a mechanism that incorporates several elements first proposed by Schwarz and Arigoni[16] (now applied to the specific reactions observed here, see below). Moreover, the carboxylic acid group formed at C2 forms the basis of one of the lactone rings in the ginkgolides[35].

## Coordinated action of CYPs from the BGC further advances ginkgolide biosynthesis via carbon skeleton rearrangement

To test whether the remaining CYPs present in the BGC could accept **7** as substrate, we used tobacco to co-express *Gb*LPS, *Gb*CYP7005C1, and *Gb*CYP7005C3 together with either *Gb*CYP7005C2, *Gb*CYP867E38, or *Gb*CYP867K1(Supplementary Data 1). Co-expression of *Gb*CYP867E38 led to a switch in the product accumulation profile from compound **7** to compound **11** (predicted formula $C_{20}H_{30}O_3$ based on accurate mass) (Fig. 3a; Table 1). By contrast, *Gb*CYP867K1 did not change the profile significantly, except for the appearance of small amounts of compound **12** (predicted formula $C_{32}H_{48}O_{12}$ based on accurate mass) (Table 1; Supplementary Figs. 3 and 6). The molecular formula of **11** suggested it is a hydroxylated derivative of **7**, while **12** seemed to be a diglycosylated diterpenoid. We then tested whether *Gb*CYP867K1 may use **11** as substrate by co-expressing it with *Gb*LPS, *Gb*CYP7005C1, *Gb*CYP7005C3, and *Gb*CYP867E38 in tobacco. Expression of this enzyme-combination resulted again in a full switch of the

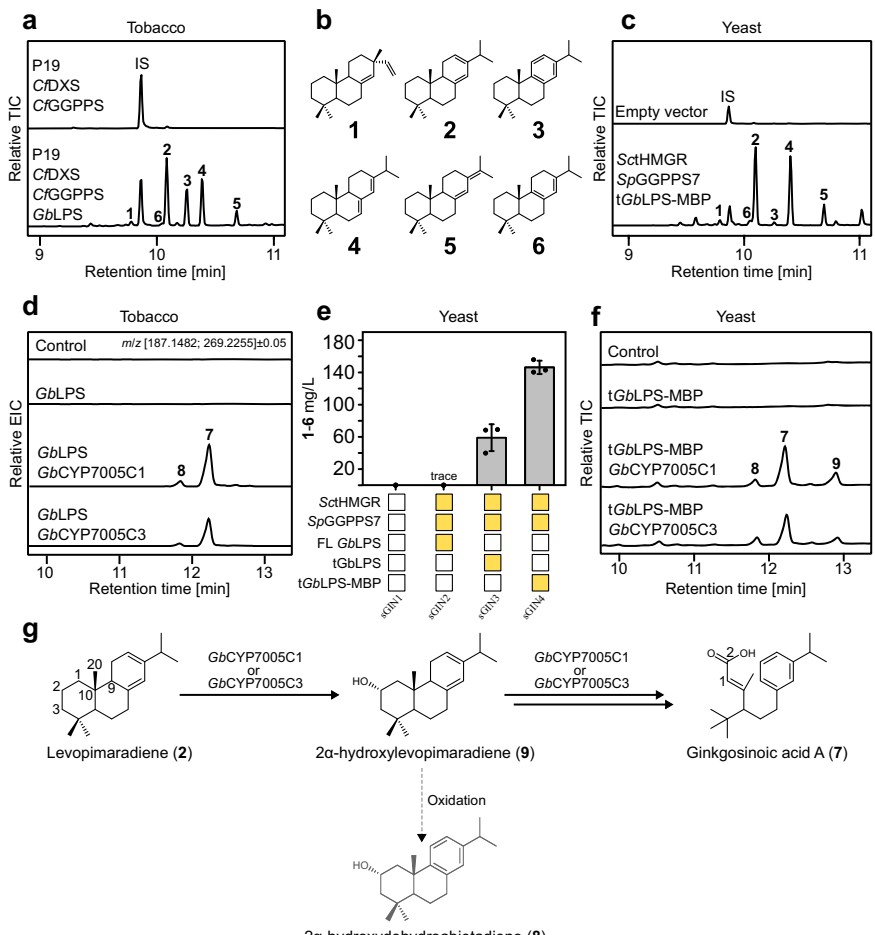

**Fig. 2 | Ginkgosinoic acid A (7) is an intermediate from levopimaradiene toward ginkgolides. a** GC–MS chromatograms of hexane extracts of *N. benthamiana* (tobacco) leaves transiently expressing *Cf*DXS (*C. forskohlii* 1-deoxy-ᴅ-xylulose 5-phosphate synthase) and *Cf*GGPPS (*C. forskohlii* geranylgeranyl pyrophosphate synthase) with or without *Gb*LPS (*G. biloba* levopimaradiene synthase). In all samples, the P19 silencing suppressor is co-expressed. IS internal standard. **b** Chemical structure of compounds **1**–**6** identified as *Gb*LPS products (Supplementary Fig. 1). **c** GC–MS chromatograms of hexane extracts of *S. cerevisiae* cells carrying an empty vector or expressing *Sc*tHMGR (a truncated version of yeast 3-hydroxy-3-methyl-glutaryl-coenzyme A reductase), *Sp*GGPPS7 (*Synechococcus* sp. geranylgeranyl pyrophosphate synthase) with a truncated version of codon-optimized *Gb*LPS fused with maltose-binding protein (MBP) for optimal expression. IS internal standard. **d** Transient co-expression of *Gb*LPS together with either *Gb*CYP7005C1 or *Gb*CYP7005C3 in tobacco leaves. The co-expression led to the accumulation of compounds **7** and **8**, as shown on extracted ion chromatograms (EICs) from LC–HRMS analysis of methanolic extracts. **e** Optimization of *Gb*LPS efficiency in yeast by truncation of the plastid transit peptide and fusion to MBP at the C-terminal end, as monitored by GC-MS-based quantification. Error bars indicate a standard deviation of $n = 3$ biological independent replicates. **f** Formation of compounds **7**–**9** in yeast cells expressing *Gb*CYP7005C1 or *Gb*CYP7005C3 together with *Gb*POR2, SpGGPPS7, SctHMGR and *Gb*LPS. The chromatograms show total ion chromatograms (TICs) from LC–HRMS analysis of methanolic yeast extracts. **g** Proposal for the reactions catalyzed by *Gb*CYP7005C1 or *Gb*CYP7005C3. The chemical structures of **7**–**9** were elucidated using NMR. Compounds **8** and **9** were established as side-product and intermediate, respectively, using feeding studies (Supplementary Fig. 5a, c, e). The numbering of carbon atoms follows the standard numbering of levopimaradiene. Source data are provided as a Source Data file.

product accumulation profile, this time from **11** into several glycosylated derivatives of a terpenoid (**13a**–**13d**), with **13d** being the predominant product (predicted formula $C_{26}H_{38}O_8$ based on accurate mass) (Fig. 3a; Table 1 and Supplementary Figs. 3, 6, 7a, 8). This suggested that *Gb*CYP867K1 catalyzed the conversion of **11** into the aglycone of **13a**–**d**, and that endogenous glycosyltransferases from *N. benthamiana* catalyzed the glycosylation reactions, possibly alleviating potential toxic effects of the aglycone. Co-expression of *Gb*CYP700C2 did not lead to any changes in the product accumulation profile in any of the co-expression combinations tested.

To avoid the extensive glycosylation observed in tobacco and obtain high amounts of products for structural characterization, we turned to yeast as an expression platform. Therefore, we integrated *Gb*CYP7005C1, *Gb*CYP7005C3, *Gb*POR2, and *Gb*CYP867E38 cDNAs into the genome of the yeast strain sGIN4 (optimized *Gb*LPS-MBP) (Supplementary Table 2). This resulted in strain sGIN11, which

produced high titers ($41 \pm 4$ mg/L) of the compound **11**, previously observed in tobacco (Fig. 3a, b). NMR spectroscopy of isolated **11** identified its structure as 12-hydroxy ginkgosinoic acid A, which we dubbed ginkgosinoic acid B (Fig. 3c; Supplementary Figs. 31–35 and Supplementary Table 6). This suggests that *Gb*CYP867E38 catalyzes the hydroxylation of ginkgosinoic acid A at position C12. Integration of *Gb*CYP867K1 cDNA into strain sGIN11 gave rise to strain sGIN13, which produced the compounds **14**–**16**. Their apparent molecular formulas suggested that they were glutathione conjugates of diterpenoids (Fig. 3b; Table 1). Purification of **14**–**16** from yeast for structural elucidation was not possible due to their hydrophilic nature and their high dilution in the yeast culture media. Consequently, we turned back to the tobacco expression system for the purification of **13d**.

To improve the yields of the heterologously produced diterpenoids in tobacco, we shifted the expression of *Gb*LPS from its natural localization in plastids to the cytosol, as this has proven to increase

**Table 1 | Products generated in this work**

| ID | $t_R$ (min) | [M + H]$^+$ (m/z) | Scan ion | Predicted formula | Error (ppm) | Putative modification | Identified compound |
|---|---|---|---|---|---|---|---|
| 7 | 12.2 | 303.2323 | 187.1482 | $C_{20}H_{30}O_2$ | −1.6 | – | Ginkgosinoic acid A |
| 8 | 11.8 | 287.2355 | 269.2255 | $C_{20}H_{30}O$ | 4.9 | – | 2α-hydroxydehydroabietadiene |
| 9 | 13.2 | 289.2521 | 289.2521 | $C_{20}H_{32}O$ | 1.8 | – | 2α-hydroxylevopimaradiene |
| 11 | 10.0 | 319.2266 | 203.1425 | $C_{20}H_{30}O_3$ | 0.4 | – | Ginkgosinoic acid B |
| 12 | 6.6 | 625.3231 | 625.3231 | $C_{32}H_{48}O_{12}$ | −1.9 | +2 Hexose | |
| 13a | 3.4 | 659.3273 | 317.2118 | $C_{32}H_{50}O_{14}$ | −0.8 | +2 Hexose | |
| 13b | 4.8 | 659.3271 | 317.2118 | $C_{32}H_{50}O_{14}$ | 0.4 | +2 Hexose | |
| 13c | 4.9 | 659.3269 | 317.2118 | $C_{32}H_{50}O_{14}$ | 0.6 | +2 Hexose | |
| 13d | 5.8 | 479.2662 | 317.2118 | $C_{26}H_{38}O_8$ | 0.4 | +Hexose | Ginkgosinoic acid C glucoside |
| 14 | 5.8 | 640.2890 | 640.2890 | $C_{30}H_{45}N_3O_{10}S$ | 1.3 | +Glutathione | |
| 15 | 5.9 | 640.2891 | 640.2891 | $C_{30}H_{45}N_3O_{10}S$ | 1.2 | +Glutathione | |
| 16 | 6.5 | 638.2737 | 638.2737 | $C_{30}H_{43}N_3O_{10}S$ | 0.7 | +Gluthathione | |
| 17 | 10.8 | 333.2062 | 333.2062 | $C_{20}H_{28}O_4$ | −0.6 | – | Ginkgosinoic acid C 1,4-benzoquinone |
| 18a | 2.4 | 657.3113 | 333.2064 | $C_{32}H_{48}O_{14}$ | 0.6 | +2 Hexose | |
| 18b | 4.3 | 657.3117 | 333.2064 | $C_{32}H_{48}O_{14}$ | 0.1 | +2 Hexose | |
| 18c | 5.2 | 495.2589 | 333.2064 | $C_{26}H_{39}O_9$ | −0.2 | +Hexose | Ginkgolactone C glucoside |
| 18d | 5.7 | 581.2597 | 333.2064 | $C_{29}H_{40}O_{12}$ | −0.7 | +Malonylglucoside | |
| 19 | 5.1 | 511.2545 | 331.1910 | $C_{26}H_{38}O_{10}$ | −4.1 | +Hexose | Ginkgolactone D glucoside |
| 20 | 5.3 | 638.2751 | 638.2751 | $C_{30}H_{43}N_3O_{10}S$ | −1.4 | +Glutathione | |
| 21 | 5.4 | 638.2742 | 638.2742 | $C_{30}H_{43}N_3O_{10}S$ | 0.9 | +Glutathione | |
| 22 | 6.0 | 636.2581 | 636.2581 | $C_{30}H_{41}N_3O_{10}S$ | 0.7 | +Glutathione | |
| 23 | 10.3 | 331.1904 | 331.1904 | $C_{20}H_{26}O_4$ | −0.1 | – | Ginkgolactone C 1,4-benzoquinone |

diterpenoid production[36]. Additionally, we co-expressed *Sp*GGPPS7 and *Sc*tHMGR in the cytosol to increase the overall pathway flux (Supplementary Fig. 7). Through a large-scale agroinfiltration experiment co-expressing *Gb*LPS, *Gb*CYP7005C1, *Gb*CYP7005C3, *Gb*CYP867E38, and *Gb*CYP867K1, we isolated **13d** and subjected it to Viscozyme L treatment to release the aglycone terpenoid (Supplementary Method 8). We expected the aglycone of **13d** to give an *m/z* of 335; instead, we observed an *m/z* of 333 (**17**), suggesting spontaneous dehydrogenation (Supplementary Figs. 3 and 9a). To understand the chemical structure of the aglycone, we subjected both **13d** and **17** to NMR spectroscopy. The structure of **13d** (Supplementary Figs. 36–40; Supplementary Table 7) suggested that it was formed from ginkgosinoic acid B via oxidation-induced migration of the large alkyl group from position C8 to position C9, which implies cleavage of the C7–C8 bond and establishment of a C7–C9 bond (Figs. 3c and 5c). This particular C–C bond shift is a key feature in the biosynthesis of ginkgolides[16]. We named **13d** ginkgosinoic acid C glucoside (Fig. 3c). The alkyl group migration may proceed via a rearrangement similar to the one hypothesized by Scharz and Arigoni[16], except that it occurs not on ferruginol, but on ginkgosinoic acid B (**11**), thus constituting a classical NIH shift[37]. Such NIH shift may occur via hydroxylation or epoxidation of the aromatic ring (for more details on the suggested mechanisms see below). The structural analysis of **17** showed that it was a 1,4-benzoquinone form of ginkgosinoic acid C (Fig. 3c; Supplementary Figs. 41–45 and Supplementary Table 8) likely to have arisen spontaneously as reported for other structurally similar diterpenoids[38] or other NIH-shifted compounds[39]. To verify that **11** serves as a substrate of *Gb*CYP867K1, we performed an in vivo feeding experiment in tobacco where we infiltrated **11** in tobacco leaves expressing solely *Gb*CYP867K1. We observed the formation of glycosides **13a**–**13d** in these leaves and not in control leaves (Supplementary Fig. 10). Accordingly, we propose that *Gb*CYP867K1 is a multifunctional enzyme that catalyzes the oxidation of ginkgosinoic acid B (**11**) followed by a carbon skeleton rearrangement occurring on a previously unsuspected pathway intermediate.

## Formation of the first lactone ring in ginkgolide biosynthesis is catalyzed by a CYP720 encoded outside the BGC

To identify further CYP enzymes that could potentially participate in ginkgolide biosynthesis, we established the CYPome of *G. biloba* by mining its genome draft as well as the available transcriptomes (Supplementary Method 1). Phylogenetic analysis of the 318 identified CYP sequences classified them into 47 CYP families and subfamilies. Of these, 36 families were shared with angiosperm species and 11 with plants from the pre-angiosperm era. We did not identify any *G. biloba*-specific CYP families, nevertheless, we observed a number of *G. biloba*-specific subfamilies like CYP7005C and several highly expanded subfamilies such as the CYP736C/E/Q (47 members), CYP76A (29 members) and CYP720B (19 members) (Supplementary Figs. 11, 12).

The biosynthesis of ginkgolides is proposed to take place in the roots of *G. biloba*[40], where *Gb*LPS is exclusively expressed[40,41]. Genes participating in the same biosynthetic pathway often share similar expression patterns[20,42,43]; therefore, we carried out a weighted correlated network analysis (WGCNA) of the nine publicly available *G. biloba* transcriptomes from the Medicinal Plant Genomics Resource (http://medicinalplantgenomics.msu.edu/). The WGCNA analysis placed the *Gb*LPS transcript in a co-expression module together with 15 *Gb*CYPs transcripts (Supplementary Table 12). Interestingly, all 5 CYPs found in the BGC are included in the *Gb*LPS-module. The remaining 10 CYPs of this module belong to subfamilies known for their involvement in gymnosperm diterpenoid biosynthesis (e.g. CYP725A[44] and CYP720B[45]) or to families, to the best of our knowledge, with no previously characterized members (e.g. CYP728Q or CYP798B). The *Gb*CYPs listed in the *Gb*LPS-module showed high expression mainly in seedlings, fibrous roots, and mature fruits while they were absent from tissues such as leaves and stems.

To test whether the *Gb*CYPs from the *Gb*LPS-module participate in ginkgolide biosynthesis, we expressed them in tobacco together with *Gb*LPS, *Gb*CYP7005C1 and *Gb*CYP7005C3, in different combinations alongside *Gb*CYP867E38 and *Gb*CYP867K1 (Supplementary Data 1). Of all the additional CYPs tested, only *Gb*CYP720B31 led to a shift in the

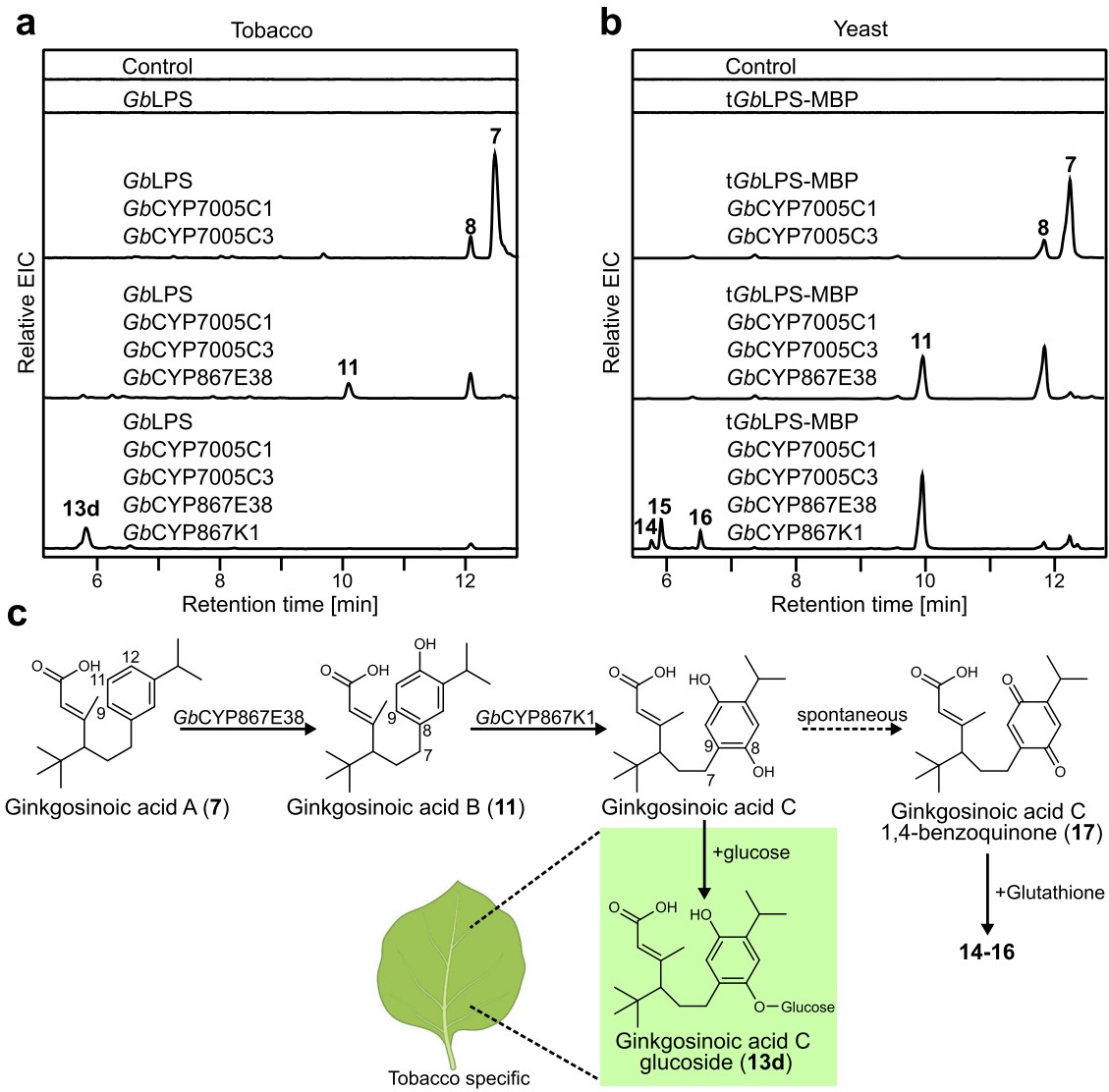

**Fig. 3 | Coordinated action of BGC-localized *Gb*CYPs (*Gb*CYP7005C1, *Gb*CYP7005C3, *Gb*CYP867E38, and *Gb*CYP867K1) advances ginkgolide biosynthesis. a** LC–HRMS analysis of tobacco leaf extracts expressing *Gb*LPS and CYP combinations as shown above each chromatogram. In all tobacco samples, *Cf*DXS (*C. forskohlii* 1-deoxy-D-xylulose 5-phosphate synthase), *Cf*GGPPS (*C. forskohlii* geranylgeranyl pyrophosphate synthase), and the P19 silencing suppressor are co-expressed. The chromatograms shown are extracted ion chromatograms (EICs, positive mode) for the *m/z* values of 187.1482; 203.1425; 317.2118; and 335.2218, each with a window of ±0.05. **b** LC–HRMS analysis of yeast terpenoid extracts from strains expressing *Sp*GGPPS7, *Sct*HMGR, *Gb*LPS, *Gb*POR2, and CYP candidate combinations as shown above each chromatogram. The chromatograms shown are

EICs (positive mode) for the *m/z* values of 187.1482; 203.1425; 638.2751; and 640.2890, each with a window of ±0.05. **c** proposed biosynthetic steps from ginkgosinoic acid A to ginkgosinoic acid C derivatives, and the responsible CYPs involved. The chemical structures of the compounds shown here (**11**, **13d**, and **17**) were elucidated using NMR spectroscopy. The putative intermediate ginkgosinoic acid C was not detected in its free form but was detected as glucoside (**13d**) in tobacco, or as glutathione conjugate (**14**–**16**) in yeast. Enzymatic in vitro removal of the glucose moiety from **13d** gave rise to **17**, a 1,4 benzoquinone derivative of ginkgosinoic acid C (Supplementary Fig. 9a). The numbering of carbon atoms follows the standard numbering of levopimaradiene. The *Nicotiana benthamiana* leaf image has been created by BioRender.com (2021).

product profile from the co-expression of *Gb*CYP7005C1, *Gb*CYP7005C3, *Gb*CYP867E38, and *Gb*CYP867K1. Instead of compounds **13a**–**d**, the addition of *Gb*CYP720B31 led to the accumulation of compounds **18a**–**d** as well as compound **19**, with **18c** being the predominant one (Fig. 4a; Table 1; Supplementary Fig. 6 and Supplementary Fig. 7a). As observed previously with **13a**–**d**, **18a**–**c** represented glycosylated variations of the same aglycone terpenoid (Supplementary Figs. 3, 8; Table 1). When *Gb*CYP720B33 was co-expressed with *Gb*LPS, it produced two minor products (**i** and **ii** in Supplementary Fig. 4a), however, no activity was observed when this enzyme was combined with any of the previously mentioned CYPs.

Expression of *Gb*CYP720B31 in yeast (Supplementary Table 2) in combination with *Gb*LPS, *Gb*CYP7005C1, *Gb*CYP7005C3,

*Gb*CYP867E38, *Gb*CYP867K1 and *Gb*POR2 resulted in the production of compounds **20**–**22**, which had apparent molecular formulas indicating glutathione conjugation (Fig. 4b; Table 1) as observed previously for compounds **14**–**16**. Therefore, to clarify the catalytic function of *Gb*CYP720B31, we turned again to the tobacco system for the isolation and structural characterization of the diterpenoid glucosides **18c** and **19** and their corresponding aglycones. Viscozyme L treatment of isolated **18c** generated the putative terpenoid aglycone **23** (Supplementary Figs. 3 and 9b), whereas isolated **19** failed to produce any detectable derivatives. The mass of **23** indicated, similarly to **17**, a dehydrogenation event in comparison to the predicted aglycone. Structural elucidation of **18c** and **19** by NMR spectroscopy revealed that the two products were glucosidic

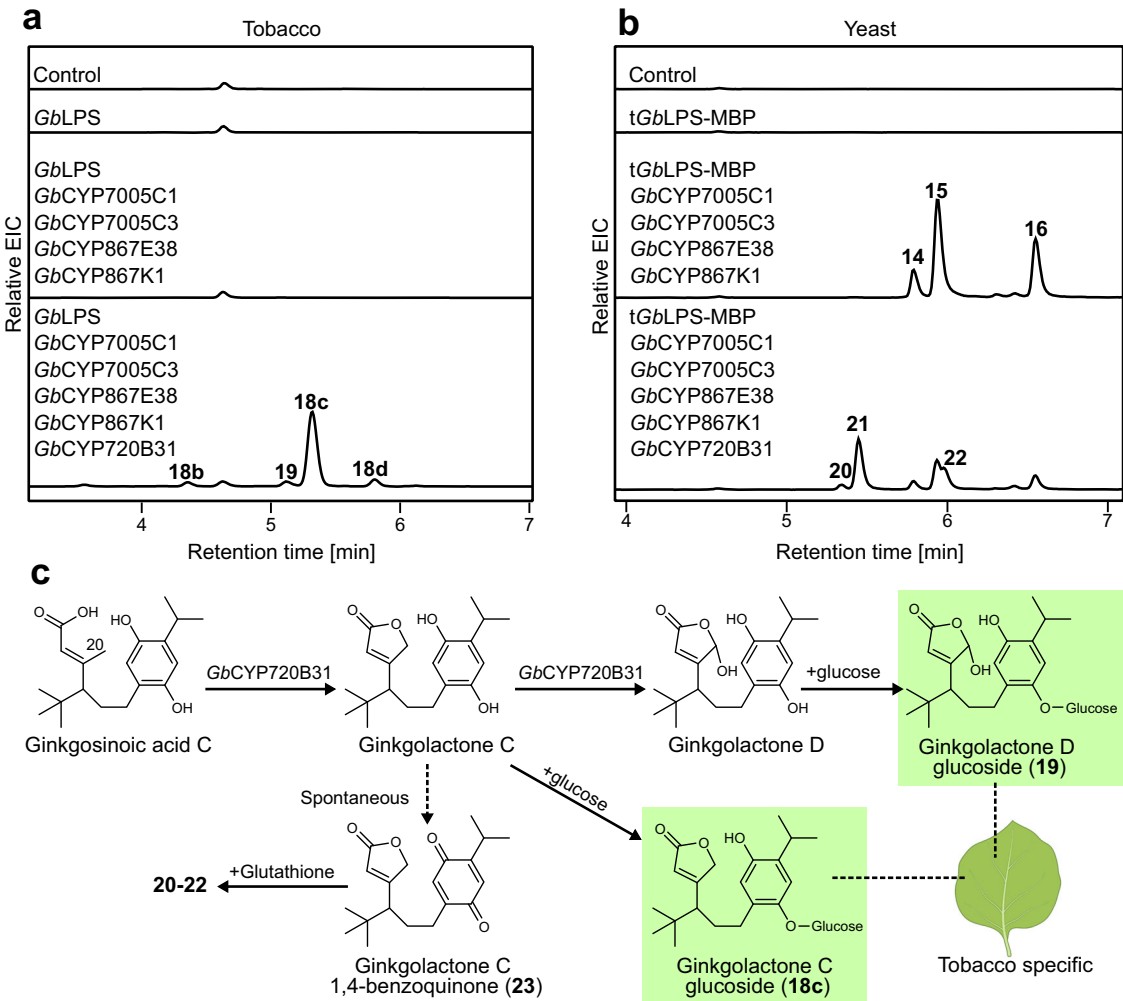

**Fig. 4 | The first lactone ring towards ginkgolide biosynthesis is generated from the activity of *Gb*CYP720B31. a** LC−HRMS analysis of tobacco leaf extracts expressing *Gb*LPS and the combinations of CYP candidates are shown in the panel. The chromatograms shown are EICs (positive mode) for the *m/z* values of 333.2062 and 331.1904, each with a window of ±0.05. **b** LC−HRMS analysis of yeast terpenoids extracts from strains expressing *Sp*GGPPS7, *Sc*tHMGR, *Gb*LPS, *Gb*POR2 and the combinations of CYP candidates shown in the panel. The chromatograms shown are extracted ion chromatograms (EICs, positive mode) for the *m/z* values 636.2581; 638.2751, and 640.2890, each with a window of ±0.05. **c** Proposed biosynthetic steps from ginkgosinoic acid C to ginkgolactone C and D derivatives, likely catalyzed by *Gb*CYP720B31. The chemical structures of the compounds shown here (**18c**, **19**, and **23**) were structurally elucidated using NMR spectroscopy. The putative intermediates ginkgolactone C and D were not detected in their free form but only as glucosides (**18c** and **19**) from tobacco, or in conjugation with glutathione (**20−22**). Enzymatic in vitro removal of the glucose moiety from **18c** gave rise to **23**, a 1,4 benzoquinone derivative of ginkgolactone A (Supplementary Fig. 9b). The numbering of carbon atoms follows the standard numbering of levopimaradiene. The *Nicotiana benthamiana* leaf image has been created by BioRender.com (2021).

lactone derivatives of ginkgosinoic acid C that varied by one hydroxyl group. We hereby name these compounds ginkgolactone C glucoside and ginkgolactone D glucoside, respectively (Fig. 4c; Supplementary Figs. 46–50, 55–59 and Supplementary Tables 9 and 10). Based on these results, we propose that *Gb*CYP720B31 hydroxylates ginkgosinoic acid C at position C20, thus enabling lactone formation via spontaneous ring closure or via further catalysis to give ginkgolactone C (Fig. 4c). In addition, *Gb*CYP720B31 is also able to hydroxylate ginkgolactone C at position C20, thus generating ginkgolactone D. In tobacco, this second hydroxylation appears to compete mainly with endogenous glucosyltransferases to give **18a–d**, whereas, in yeast, it appears to compete mainly with dehydrogenation and subsequent glutathionylation to give **20–22** (Fig. 4c).

Compound **23** (similarly to **17**) was shown by NMR spectroscopy to be a 1,4-benzoquinone derivative of ginkgolactone C (Supplementary Figs. 9b, 51–54; Supplementary Table 11). Examination of tobacco extracts from leaf tissue expressing *Gb*LPS, *Gb*CYP7005C1, *Gb*POR2, *Gb*CYP7005C3, *Gb*CYP867E38, *Gb*CYP867K1 with or without

*Gb*CYP720B31 showed that it was possible to detect compounds **14–16** and **20–22** (previously only noted in yeast) from the respective gene combinations (Supplementary Fig. 13). This demonstrates the highly reactive nature of the 1,4-benzoquinones **17** and **23** resulting in adduct formation with glutathione, independent of the expression host. When **17** and **23** were incubated in vitro with glutathione, glutathione adducts identical to those observed in yeast and tobacco were formed (Supplementary Figs. 14, 15). This indicates that the conjugation to glutathione likely happened spontaneously, however, we cannot rule out the possibility that endogenous enzymes catalyzed the conjugation in each of the host systems.

## Discussion

In this study, we used a BGC-guided mining approach in combination with gene coexpression analysis to uncover a number of cytochrome P450s able to catalyze reactions towards the synthesis of ginkgolides (Fig. 5). With exception of *Gb*CYP720B31, the identified enzymes belong to CYP families absent in modern seed plants and, to the best of our knowledge, with no previously functionally characterized

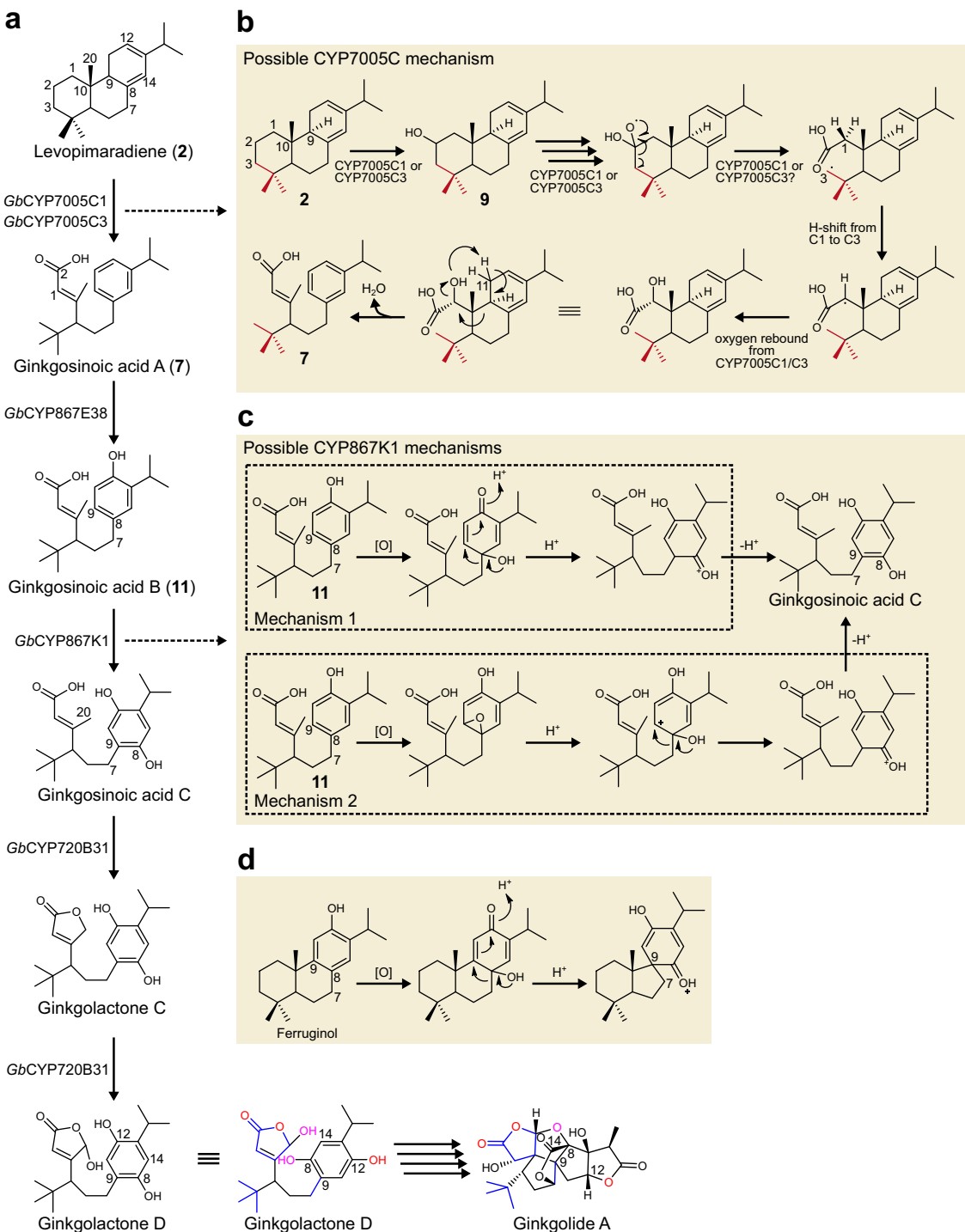

**Fig. 5 | Proposed initial steps in the conversion of levopimaradiene (2) to ginkgolides in *G. biloba*. a** The enzymes identified in this work, the suggested activity, and the identified pathway intermediates. The colors depicted in ginkgolactone D highlight the structural characteristics that match the final configuration in ginkgolide A, also colored correspondingly. Ginkgolactone D is shown in two different conformations for easier visualization of its conversion into ginkgolide A. **b** Possible reaction mechanism of the conversion of levopimaradiene (2) to ginkgosinoic acid A (7), the first suggested intermediate in the biosynthesis of ginkgolides. The proposal includes a radical ring opening followed by a similar hydrogen shift (H-shift) as the one proposed by Schwarz and Arigoni[16]. Oxygen rebound to quench the free radical (now at position C1) leads to an unstable intermediate that undergoes a second ring opening concomitant with water elimination. **c** Possible reaction mechanisms affording the **C7–C8** to **C7–C9** bond-shift in ginkgolide biosynthesis. *Mechanism 1* is involving a dienone-phenol rearrangement starting with ginkgosinoic acid B. The proposed mechanism amounts to a classical NIH shift. *Mechanism 2* depicts an alternative mechanism for the NIH shift, starting with epoxidation of ginkgosinoic acid B followed by acid-catalyzed opening of the epoxide. The localized carbocation is then stabilized by alkyl group migration, ultimately leading to ginkgosinoic acid C. **d** shows the relevant mechanism as proposed by Schwarz and Arigoni[16], involving a dienone-phenol rearrangement of ferruginol towards ginkgolides.

members. Our results indicate levopimaradiene (**2**) as the precursor of all ginkgolides and show that it is converted by *Gb*CYP7005C1 or *Gb*CYP7005C3 to ginkgosinoic acid A (**7**) via two extraordinary C−C cleavage events, one of them leading to the characteristic *tert*-butyl group present in both ginkgolides and bilobalide. Next, *Gb*CYP867E38 hydroxylates ginkgosinoic acid A to give ginkgosinoic acid B (**11**). Ginkgosinoic acid B is then converted by *Gb*CYP867K1 to ginkgosinoic acid C (**13d** aglycone) through aromatic ring oxidation leading to alkyl chain migration (NIH shift). Finally, ginkgosinoic acid C is converted by *Gb*CYP720B31 to ginkgolactones C (**18c** aglycone) and D (**19** aglycone). Our proposed reaction mechanisms (Fig. 5b, c) incorporate many of the observations made by Schwarz and Arigoni[16] but the resulting pathway proposal is notably different and defines a previously unsuspected early route for ginkgolide biosynthesis.

Our work shows that the identification of BGCs, as well as monitoring gene co-expression patterns, can facilitate the discovery and elucidation of highly complex plant pathways from early diverging plants with large genomes such as the ginkgo tree. Although a variety of BGCs has been identified in plants[18,19,46,47], the principles behind gene cluster assembly remain a matter of debate, and so do the evolutionary positive selection pressure for their assembly or the negative selection regarding their dissociation[18,47]. Although the identified BGC does not include the entire set of ginkgolide biosynthetic genes, its presence suggests a strong selection pressure for assembly and preservation[47], most likely as a mechanism for reassuring the efficient biosynthesis of ginkgolides. This consequently demonstrates the evolutionary importance of ginkgolides in the physiology, fitness, and survival of the *G. biloba* tree[2]. It is likely that the assembly and preservation of this BGC were triggered by the requirement of tight and coordinated regulation of the expression of its genes. This is supported by the fact that all CYPs found in the BGC share the same expression pattern with *Gb*LPS and are included in the same co-expression module composed of genes mainly expressed in the roots. In turn, the requirement for tight co-expression may respond to a need for preventing the accumulation of unstable or toxic pathway intermediates[46]. Interestingly, we observed that the products of the heterologously expressed *Gb*CYPs downstream of ginkgosinoic acid B (**11**) were modified by conjugation with glutathione or by glycosylation in both production hosts, possibly indicating instability and/or toxicity. In addition, ginkgosinoic acid C, ginkgolactone C and ginkgolactone D, are converted spontaneously to their 1,4-benzoquinone derivatives, further supporting this idea. None of these ginkgolide intermediates have been identified in ginkgo tissues[16], suggesting effective metabolic channeling. Identification of the remaining steps of the pathway will show whether the missing biosynthetic genes are organized in different BGCs and whether they share similar expression profiles with the already identified genes.

To the best of our knowledge, the CYPs identified in the reported BGC are classified into families with no previously functionally characterized members[23] like the *Gb*CYP7005Cs which belong to the CYP85 clan (Supplementary Fig. 12). Prior to the release of the ginkgo genome, the CYP7005 family was considered specific to ferns[22], with the fern CYPs hosted within the CYP7005A subfamily. Members of this subfamily share approximately 40% sequence homology with their ginkgo counterparts. Ferns are primitive plants, which evolved before the speciation of seed plants (gymnosperms and angiosperms)[48,49]. Thus, the CYP7005 family corresponds to ancient CYPs absent in modern plants, the only exception being *G. biloba*. It is safe to assume that the almost complete extinction of the Ginkgophyta lineage (the only lineage that carried this enzyme family beyond ferns)[4,50] likely contributed to halting the evolution of this enzyme family. To the best of our knowledge, no other members of the CYP7005 family have been characterized so far.

Some of the reactions catalyzed by the identified ginkgo CYPs are complex. In particular, the reaction catalyzed by *Gb*CYP7005C1 or

*Gb*CYP7005C3 involves C−C bond cleavage at two different positions in levopimaradiene, with the first cleavage likely following a radical mechanism (Fig. 5b). Particularly, this cleavage does not leave a scar where the radical is originally created upon cleavage (C3), thus enabling the formation of an intact *tert*-butyl group. The scarless cleavage can be explained by the hydrogen shift mechanism initially proposed by Schwarz and Arigoni[16], where a hydrogen atom migrates from the nearby C1 position, thus leaving a radical at C1 instead. The radical can be quenched by oxygen rebound from the CYP, giving a presumably unstable intermediate that can rearrange by simultaneous aromatization, dehydration, and heterolytic C−C bond cleavage (Fig. 5b). Whether *Gb*CYP7005C1 and *Gb*CYP7005C3 are directly involved in catalyzing all of these mechanistic steps remains to be shown; however, both CYPs can safely be considered multifunctional. Multifunctional CYPs are common in plant metabolism, for example, the CYP88A (or ent-kaurenoic acid oxidase, KAO) in gibberellin biosynthesis. KAO catalyzes the conversion of ent-kaurenoic acid to $GA_{12}$ in three steps[51,52], with no escape of any intermediates from its active site. First, ent-kaurenoic acid is oxidized via a stereospecific hydroxylation at C-7 to form 7β-hydroxy-ent-kaurenoic acid, in the next step the ring B contracts from 6 to 5 carbon atoms via the migration of C7-C8 to C6-C7 bond, followed by the formation of $GA_{12}$-aldehyde. In the final step, $GA_{12}$-aldehyde is oxidized to $GA_{12}$. The discovery of *Gb*CYP7005C1 and *Gb*CYP7005C3 can further contribute to the understanding of CYPs multifunctionality on the structural level, for example, assisted by research on crystal structures[53] or molecular dynamics simulations[54].

The reaction catalyzed by *Gb*CYP867K1 is also unique, particularly because biosynthetic NIH shifts are rare, and those involving alkyl chain migration, even more so. The NIH shift was first reported by researchers from the US National Institutes of Health (NIH), who observed the migration of hydrogen isotopes upon aromatic hydroxylation of xenobiotic compounds as part of mammalian detoxification systems[55]. A limited number of examples have emerged where the migrating group is not a hydrogen atom, but an alkyl substituent. These examples are mostly restricted to degradative pathways, such as in the catabolism of L-tyrosine[56] or in the biodegradation of a-tertiary nonylphenols by *Sphingobium xenophagum*[39]. We are only aware of one other case of biosynthetic NIH-shift-mediated migration of an alkyl group. It occurs in the biosynthesis of pseudoisoeugenol and derivatives in anis plants[57]; however, the responsible enzyme remains unknown. Using substrate feeding in the absence of other pathway enzymes, we show unequivocally that *Gb*CYP867K1 catalyzes the conversion of ginkgosinoic acid B to ginkgosinoic acid C (Supplementary Fig. 10), making it a biosynthetic enzyme to catalyze such alkyl group migration. Moreover, the carbon skeleton rearrangement caused by this migration was previously proposed to occur on ferruginol or a related compound (Fig. 5d). The knowledge that ginkgosinoic acid B is the subject of the rearrangement firmly establishes an updated hypothesis for the early steps in ginkgolide biosynthesis.

*Gb*CYP867E38 and *Gb*CYP867K1 belong to the CYP867 family. Members of this family have been identified in cycads and conifers but not in flowering plants (angiosperms). Again, to the best of our knowledge, there are no previously reported characterized members from this family[23].

The final CYP found in this work to participate in ginkgolide biosynthesis is *Gb*CYP720B31, which is not encoded in the BGC, but is part of the *Gb*LPS co-expression module. Enzymes of the CYP720 family belong to the CYP85 clan. Although not gymnosperm-specific, members of the CYP720 family are absent from monocots, and they are represented by very few members in dicots[58]. The family is highly expanded in gymnosperms and, specifically, the CYP720B subfamily has been associated with the

biosynthesis of oleoresin terpenoids[45] produced after stress elicitation. The typical activity of CYP720B enzymes is the oxygenation of C18 in diterpene hydrocarbons with tricyclic backbones, including levopimaradiene and dehydroabietadiene[59]. CYP720B enzymes have been reported to show substrate promiscuity[45], but our study showed that a CYP720B can accept a substrate like ginkgosinoic acid C, which is devoid of a tricyclic skeleton. In our efforts to identify ginkgolide biosynthetic genes in the vicinity of *GbCYP720B31*, located in chromosome 4, we identified a single CYP-encoding gene, *GbCYP720B49*. This CYP had already been tested as part of the *Gb*LPS co-expression module (Supplementary Table 12). Co-expression of *Gb*CYP720B49 with the already known genes did not lead to any changes in the product profile. Nevertheless, this does not rule out participation in later steps of the ginkgolide pathway.

Future strategies that could assist the identification of the next steps in ginkgolide biosynthesis include the identification of additional BGCs as well as the discovery of co-expression modules, especially under environmental stimuli that induce the biosynthesis of ginkgolides (e.g. jasmonic acid[60]). Further strategies include the testing of additional CYPs from the families found to be involved in the ginkgolide pathway (e.g. CYP867) and the characterization of the transcription factor encoded in the BGC, including the genes that it might regulate. Undoubtedly, it would be of great interest to manipulate the expression of specific gene candidates in the ginkgo tree to study the effects of knocking them down/out. Nevertheless, such approaches are not possible in this living fossil as we have no means of genetically manipulating it yet.

The work reported here represents proof that the identification of BGCs, in combination with gene co-expression studies, represents a valuable tool for the elucidation of complex biosynthetic pathways, e.g. for ginkgolides. The CYPs identified and characterized in this work hold a particular evolutionary status among plant CYPs, which reinforces the characterization of ginkgo as a 'living fossil' not only because of its morphological stasis from the Jurassic period to today but also due to biochemical and molecular data.

## Methods

### Co-expression analysis, hub-gene identification, and network analysis

Co-expression analysis was performed using the weighted gene correlation network analysis (WGCNA) package in R under the guidelines of the published tutorials[61,62]. Nine *G. biloba* transcriptome datasets were downloaded from www.medicinalplantgenomics.msu.edu and log2-normalized FPKM values were imported into R. Only genes with FPKM value of >3 in at least 2 samples were included in the analysis. Moreover, genes were tested for their co-efficient of variation (COV), and low variant genes (COV < 0.2) were removed from the analysis as well, leaving a dataset including 8512 genes for the co-expression analysis. Further WGCNA analysis was performed using default settings with minor changes. We used a soft power of 13 for the calculation of network adjacency of gene counts and the topological overlap matrix (TOM). Subsequent clustering of modules was completed with a minimum module size of at least 30 genes and an automatic tree cut of 0.989 for the adaptive branch pruning and a deep split of 2. In order to combine modules that are too close, we merged close modules at a maximum dissimilarity of 0.2, ending up with 28 modules.

### Genome analysis and identification of a gene cluster

An assembled *G. biloba* genome was downloaded from the GigaScience Database (http://gigadb.org/dataset/view/id/100613/Sample_sort/genbank_name) along with the gene annotation file (GFF file). CLC Main Workbench 20 (Qiagen, Denmark) was used to annotate the genome sequences and for BLAST searching gene locations.

### RNA isolation, cDNA synthesis, and cloning of candidate transcripts

*Ginkgo biloba* plants were bought from a plant nursery. Approximately 100 mg of liquid nitrogen frozen and ground *G. biloba* tissue was used to extract RNA using the Spectrum™ Plant Total RNA kit (Sigma-Aldrich, Germany) from either leaf or fibrous root tissue following the provided instructions. The quality of the RNA was analyzed by Bioanalyzer 2100 (Agilent, USA). cDNA was synthesized using the SuperScript® IV First-Strand Synthesis System (Thermo Fisher Scientific, USA) using the standard protocol and oligo(dT)$_{20}$ primers. cDNAs were PCR amplified using gene-specific primers (Supplementary Table 13) and PCR products were purified by gel-purification (E.Z.N.A® Gel Extraction Kit, Omega Biotech, USA). The pJET1.2 cloning kit (Thermo Fisher Scientific, USA) was used to clone the blunt-end PCR products in a 20 μL reaction following standard protocol. A total of 5 μL pJET1.2 cloning reaction was used to transform 50 μL E. Cloni® 10G Competent Cells (Lucigen, USA) by standard protocol. Plasmids with gene candidates were sequenced (Macrogen, South Korea) after plasmid purification (E.Z.N.A® Plasmid Mini Kit I, Omega Biotech, USA).

### USER cloning of *Nicotiana benthamiana* and *Saccharomyces cerevisiae* constructs

Constructs for transient expression in *Nicotiana benthamiana* (tobacco) were generated using the pLIFE33 vector (Supplementary Table 14). cDNAs were cloned using the Uracil-Specific-Excision-Reaction (USER) method[63] with USER cloning-specific primers (Supplementary Table 15). Vectors for expression by genomic integration in *Saccharomyces cerevisiae* (Supplementary Table 16; Supplementary Table 17) were cloned as follows: Promoter fragments for either single or dual gene constructs, as well as gene fragments, were amplified using USER cloning specific primers (Supplementary Tables 18 and 19, Supplementary Data 2). 4–8 μg of DNA vectors for USER cloning were linearized in 50 μL reactions firstly by AsiSI (New England Biolabs, USA) overnight at room temperature followed by gel purification and nicking by Nb.BmsI (New England Biolabs, USA) overnight at room temperature followed by 20 min inactivation at 80 °C. Single or dual gene constructs were USER cloned in 10 μL reactions with 1 μL CutSmart® buffer (New England Biolabs, USA), a total of 7 μL DNA insert (Promoter and gene fragments), 1 μL linearized vector backbone, and 1 μL USER™ enzyme (New England Biolabs, USA). All USER reactions were carried out in PCR-strips at 37 °C for 20 min, 20 °C for 20 min followed by 10 °C for 10 min. 50 μL E. Cloni® 10G Competent Cells (Lucigen, USA) were added to each USER reaction on ice, and transformation was carried out following standard protocol. *G. biloba* selected cDNAs have been codon optimized for yeast expression. The relevant sequences are presented in Supplementary Data 3.

### Bacteria and yeast medium compositions and cultivation

*Escherichia coli* was cultivated in LB medium (10 g/L tryptone, 5 g/L yeast extract, 5 g/L NaCl) with either 50 μg/mL carbenicillin or 50 μg/mL kanamycin. *Agrobacterium tumefaciens* strain AGL-1-GV3850 was cultivated in YEP medium (10 g/L bacto-tryptone, 5 g/L yeast extract, 10 g/L NaCl) containing 50 μg/mL kanamycin, 25 μg/mL rifampicin, and 50 μg/mL carbenicillin. Yeast strains without the URA3 marker were cultivated and maintained in YPD medium (10 g/L yeast extract, 20 g/L bacto-peptone and 2% glucose). Strains harboring the URA3 marker were maintained in a synthetic-complete (SC) medium without uracil (1.92 g/L synthetic complete drop-out powder without uracil (Sigma-Aldrich, Germany), 6.7 g/L yeast nitrogen base (Sigma-Aldrich, Germany) and 2% glucose). 5-Fluoroorotic Acid (5-FOA) plates were used to select URA3 looped-out clones (0.1% 5-Fluoroorotic Acid, 1.92 g/L synthetic complete drop-out powder without uracil (Sigma-Aldrich, Germany), 6.7 g/L yeast nitrogen base (Sigma-Aldrich, Germany), 50 mg/L uracil and 2% glucose).

Yeast strains (Supplementary Table 2) cultivated for the production of diterpenoids were grown in a fed-batch-like mineral medium with the EnPump 200 slow glucose release reagent (Enpresso, Germany). 1 L medium consisted of 480 mL salt mix (85.2 g/L MES monohydrate, 15.4 g/L $(NH_4)_2SO_4$, 8.4 g/L citric acid monohydrate, 6.6 g/L KCl, 6 g/L $K_2HPO_4$, 6 g/L $MgSO_4$ heptahydrate, 6 g/L NaCl (pH 6.4)), 390 mL 100 g/L EnPump 200 substrate (Enpresso, Germany) in a phosphate buffer (3.4 g/L $NaH_2PO_4$, and 20.2 g/L $Na_2HPO_4$, (pH 7)), 9 mL $CaCl_2$ solution (112 g/L $CaCl_2$ dihydrate), 10 mL vitamin mix (0.64 g/L D-biotin, 3 g/L Nicotinic acid, 10 g/L Thiamin HCl, 4 g/L D-pantothenic acid hemicalcium salt, 8 g/L myo-inositol, 2 g/L pyridoxine HCl), 10 mL microelements (6.7 g/L Titriplex III, 6.7 g/L $(NH_4)_2Fe(SO_4)_2$ hexahydrate, 0.55 g/L $CuSO_4$ pentahydrate, 2 g/L $ZnSO_4$ heptahydrate, and $MnSO_4$ monohydrate), 1 mL trace elements (1.25 g/L $NiSO_4$ hexahydrate, 1.25 g/L $CoCl_2$ hexahydrate 1.25 g/L, boric acid, 1.25 g/L KI, and 1.25 g/L $Na_2MoO_4$ dihydrate) and 6 mL Reagent A (Enpresso, Germany).

Yeast strains for the production of diterpenoids were grown in 2.2 mL 96 deep well plates with round bottoms with air-penetrable metal lids (EnzyScreen, The Netherlands) containing 500 µL SC-URA medium overnight at 360 RPM (2.5 orbit cast) at 30 °C. 50 µL pre-culture were transferred to new plates containing 500 µL of the above-described fed-batch-like medium at 360 RPM (2.5 orbit cast), 30 °C for 72 h before extracting metabolites.

Yeast strains were fed **8** and **9** in 25 mL shake flask cultures by firstly growing overnight cultures in SC-URA medium in growth tubes. 200 µL pre-cultures were added to 2 mL of the fed-batch-like medium in 25 mL shake flasks, and substrates were added by adding them dissolved in 100 µL DMSO (unknown concentrations). Strains were grown at 150 RPM (2.5 orbit cast), 30 °C for 72 h before extracting metabolites.

### Transient expression of candidate genes in *Nicotiana benthamiana*
USER cloned constructs for *N. benthamiana* (tobacco) transient expression were transformed into *A. tumefaciens* strain AGL-1- GV385[26]. The $OD_{600}$ of overnight cultures of *A. tumefaciens* strains were normalized to $OD_{600}$ of 1 and strains carrying genes to be transiently co-expressed were mixed in equal volumes. 4–6 weeks old tobacco plants grown in greenhouse conditions (16 h light at 20 °C, 8 h dark at 19 °C) were used for infiltration. Two leaves per plant and three plants for each enzyme combination were infiltrated by syringe, and plants were kept at greenhouse conditions (16 h light at 20 °C, 8 h dark at 19 °C) for 7 days. 2 cm diameter leaf discs were excised from the infiltrated areas and two leaf discs were combined from each combination in glass HPCL vials. Leaf discs were frozen at −71 °C and crushed using a plastic pestle. Metabolites were extracted as described below. Feeding of **7** and **11** was conducted by firstly infiltrating *N. benthamiana* plants with *A. tumefaciens* strains harboring plasmids with genes of interest and letting plants in greenhouse conditions for 72 h. Substrates were dissolved in 10% methanol (unknown concentrations), and leaves were infiltrated in the same area as the *A. tumefaciens* infiltrations. Plants were returned to the greenhouse for another 72 h before extracting as described below.

### Genetic engineering of *Saccharomyces cerevisiae*
The *S. cerevisiae* strain NCYC3608 was purchased from the National Collection of Yeast Cultures (NCYC, United Kingdom) and was used as the base strain for all engineering (Genotype S288C, MATα, *SUC2, gal2, mal2, mel,flo1, flo8-1, ho, bio1, bio6, ura3Δ*). Cassettes containing genes to be inserted into the genome were released from the Assembler plasmids by NotI (New England Biolabs, USA) digestion prior to transformation. Generally, 10 µg plasmid were digested in a 50 µL volume for 3 h with NotI at 37 °C followed by inactivation at 65 °C for 20 min. NotI released cassette combinations to be integrated into

yeast were combined in concentrations of 0.5–1 µg per cassette in sterile PCR strips. Yeast cells were made competent by growing cultures in 50 mL YPD in 250 mL Erlenmeyer flasks (180 RPM at 30 °C) until $OD_{600}$ 0.6–0.9 and collecting cells by centrifugation. Cells were washed twice with 45 mL Milli-Q water and 100 µL Milli-Q water was used to re-suspend the cells. 10 µL cell suspensions were used for each transformation. The transformation was carried out using a lithium-acetate protocol[64]. The URA3 marker was looped out by plating cells firstly on YPD agar, picking colonies after 3 days and plating these on 0.1% 5-Fluoroorotic acid (5-FOA). Colonies appearing on 5-FOA plates were plated on YPD and SC-URA to confirm which colonies were URA3 negative.

### Extraction of metabolites for LC−HRMS and GC−MS analysis
Metabolites from frozen and ground tobacco leaf discs were extracted for GC−MS analysis with analytical grade *n*-hexane (Sigma-Aldrich, Germany): 1 mL *n*-hexane spiked with 10-PPM (parts per million) 1-eicosene (Sigma-Aldrich, Germany) was added followed by vortexing and shaking at 200 RPM for 1 h. Cell debris was pelleted by centrifugation and 200 µL *n*-hexane extracts were transferred to new vials with inserts ready for analysis. Metabolites for liquid-chromatography−high-resolution-mass-spectrometry (LC−HRMS) were extracted from tobacco leaves with 1 mL HPLC-grade 100% methanol (Sigma-Aldrich, Germany) followed by vortexing and shaking at 200 RPM for 1 h. 150 µL methanol extracts were transferred to 0.22 µM filter plates and filtered before being transferred to new vials with inserts ready for LC−HRMS analysis. Samples for quantitative analysis contained 6.25 PPM andrographolide (Carbosynth, United Kingdom) as an internal standard to calculate normalized yield (extracted peak area/extracted peak area of the internal standard).

Yeast samples were extracted for GC−MS analysis by transferring 300 µL broth (cells and medium) to glass vials and adding 600 µL *n*-hexane spiked with 10-PPM 1-eicosene. Samples were vortexed 30 s followed by 1 h shaking at 300 RPM. The *n*-hexane-phase was transferred to new vials after centrifugation, ready for analysis.

Yeast metabolites for LC−HRMS analysis were extracted as follows: 100 µL broth (yeast cells and medium) were added to glass vials and added 400 µL HPLC-grade 100% methanol (Sigma-Aldrich, Germany) spiked with 6.5 PPM andrographolide as an internal standard. Samples were vortexed and left shaking at 200 RPM for 1 h. A total of 150 µL methanol extracts were transferred to 0.22 µM filter plates and filtered before being transferred to new vials with inserts ready for LC−HRMS analysis.

### Analysis of terpenoids by GC−MS and LC−HRMS
Gas-chromatography−mass-spectrometry (GC−MS) was carried out using the Shimadzu GCMS-QP2010 Ultra system equipped with an Agilent HP-5MS column (30 m × 0.25 mm i.d., 0.25 µm film thickness). The injection volume was set to 1 µL and the injection temperature at 250 °C. The GC program was as follows: 80 °C for 2 min, ramp at a rate of 30 °C min$^{-1}$ to 170 °C and held for 3 min, ramp at a rate of 30 °C min$^{-1}$ to 280 °C and held for 3 min. The total run time was 14.67 min. The ion source temperature of the mass spectrometer (MS) was set to 250 °C and spectra were recorded from *m/z* 50 to *m/z* 400. Sandaracopimaradiene (**1**), levopimaradiene (**2**), dehydroabietadiene (**3**), abietadiene (**4**), neoabietadiene (**5**), and palustradiene (**6**) were identified based on authentic standards, retention time and fragmentation patterns in comparison to reference spectra in databases (Wiley Registry of Mass Spectral Data, 8th Edition, July 2006, John Wiley & Sons, ISBN: 978-0-470-04785-9) and previously reported spectra. Titers of **1**–**5** and **6** from yeast strains were measured in 1-eicosene (Sigma-Aldrich, Germany) equivalents prepared as a standard curve.

Liquid-chromatography−high-resolution-mass-spectrometry (LC−HRMS) analysis was performed on the Dionex UltiMate® 3000 Quaternary Rapid Separation UHPLC focused system (Thermo Fisher

Scientific, Germering, Germany) equipped with a Phenomenex Kinetex XB-C18 column (100 mm × 2.1 mm i.d., 1.7 μm particle size, 100 Å pore size) (Phenomenex, Inc., Torrance, CA, USA). The column was operated at 40 °C, and the flow rate was maintained at 0.3 mL/min. The mobile phases were water (A) and 100% acetonitrile (B), both acidified with 0.05% formic acid. Separations were performed using the following gradient profile: 0 min, 20% B; 11 min, 80% B; 21 min, 90% B; 22 min, 100% B; 27 min, 100% B; 28 min, 20% B. The column outlet was connected to a Bruker Daltonics Compact QqTOF mass spectrometer equipped with an electrospray ionization (ESI) interface (Bruker Daltonics, Bremen, Germany). Mass spectra were acquired in positive ion mode, using a capillary voltage of 4000 V, an end plate offset of −500 V, a drying temperature of 220 °C, a nebulizer pressure of 2.0 bar, and a drying gas flow of 8 L min$^{-1}$. Sodium formate solution (internal standard) was injected at the beginning of each chromatographic run, and the LC−HRMS raw data were calibrated against these sodium clusters using the Data Analysis 4.3 (Bruker Daltonics) software program. Data from the Dionex UltiMate® 3000 Quaternary Rapid Separation UHPLC focused system and Bruker Daltonics Compact qTOF mass spectrometer was collected using ThermoFisher Chromeleon 6.80 software and Bruker Hystar 3.2.

### Isolation and purification of CYP products and NMR analysis
See Supplementary Method 5–7 for instrumentation settings, analysis, and isolation of products for analysis.

### Reproducibility
All *N. benthamiana* experiments were carried out in biological triplicates (three separate plants) and in multiple independent experiments. All *S. cerevisiae* strains were generated in biological replicates (three independent transformants) and data was obtained through multiple independent experiments.

### Reporting summary
Further information on research design is available in the Nature Research Reporting Summary linked to this article.

## Data availability
*Ginkgo biloba* cDNA sequences identified and sequenced here have been deposited in GenBank under the accession numbers: ON759313, ON759314, ON759315, ON759316, ON759317, ON759318, ON759319, ON759320, ON759321, ON759322, ON759323, ON759324, ON759325, ON759326, ON759327, ON759328, and ON759329. Source data are provided with this paper.

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

## Acknowledgements

We would like to thank David Pattison, Isabel Ovejero Lopez, and Jack Olsen, all at the University of Copenhagen, for their assistance in running analytical instruments. This work is financially supported by the Danish Independent Research Council—Technology and Production Sciences ("From Prehistory to the Future: Expanding the potential of *Ginkgo biloba*", Grant number 8022-00254B, awarded to I.P.) and the Lundbeck Foundation ("Brewing diterpenoids", Grant number R199-2015-450, awarded to B.L.M.). BioRender.com was used to generate part of the images shown in Figs. 1a, 3c, and 4c.

## Author contributions

V.F., I.P. conceived and designed the experiments. V.F. conducted all cloning, engineering of *N. benthamiana*, engineering of *Saccharomyces cerevisiae* and analyzed chromatography data. D.L. and D.S. isolated products from *Saccharomyces cerevisiae* or *Nicotiana benthamiana* and analyzed NMR data. F.G.-F. provided theoretical and experimental

advice on enzyme reactions and mechanisms. R.L. performed the co-expression analysis. D.R.N. performed the CYPome analysis. I.P. and B.L.M. provided mentoring, V.F., I.P., S.C.K., F.G.-F., and B.L.M. wrote the manuscript.

## Competing interests

The authors declare no competing interests.
