## [Peer Review File · Nature Communications]

A gene cluster in *Ginkgo biloba* encodes unique multifunctional cytochrome P450s that initiate ginkgolide biosynthesisReviewers' Comments:

Reviewer #1:

Remarks to the Author:

This manuscript describes key insights into biosynthesis of the structurally fascinating and pharmaceutically important ginkgolide diterpenoids. In particular, building on the previous identification of the initiating diterpene cyclase, levopimaradiene synthase (GbLS), a biosynthetic gene cluster (BGC) was identified in the publicly available genome sequence for *Ginkgo biloba*. In addition to GbLS, this BGC contains five cytochromes P450 (CYPs), at least three of which were shown here to catalyze reactions relevant to ginkgolide biosynthesis. First, the almost identical CYP7005C1 and CYP7005C3 both catalyze a mechanistically unusual and fascinating transformation that involves dual scission of the A/B decalin rings, conversion of C2 from the A ring to a carboxylic acid moiety and formation of 1(10) double bond, as well as aromatization of the remaining C ring, forming what the authors term ginkgolic acid A. Two C2-hydroxylated derivatives were observed, 2-hydroxy-levopimaradiene and the further oxidized 2-hydroxy-dehydroabietadiene, with aromatized C ring. While it seems certain that the first is an intermediate, the status of the second is perhaps less certain than indicated by the authors. Specifically, it might be speculated that cleavage between C9-C10 (B ring), if not also C2-C3 (A ring), could be coupled to C ring aromatization. An experiment is reported indicating dehydroabietadiene is not a substrate for these CYPs. Given that sufficient amounts of the two putative intermediates were isolated for NMR analysis, it would be of interest to determine if these can serve as substrates. Given that either CYP7005C1 or CYP7005C3 catalyzes this transformation to ginkgolic acid A, it is unclear why all subsequent experiments utilize recombinant co-expression of both (indeed, Supplemental Fig. 3 suggests this decreases yield, at least in tobacco, while no data is presented for yeast; as a minor correction the peak in Supplemental Fig. 11 probably should be labeled as '7' rather than '6' and the double bonds for miltiradiene in Supplemental Data Fig. 1 are incorrectly positioned, also please describe construction of the GbLS-MBP fusion in the supplemental methods). Regardless, it was then found that the two CYP867 family members in the BGC catalyze hydroxylation at C12 (CYP867E38) to form ginkgolic acid B, and, subsequently, the mechanistically interesting 1,2-migration of the cleaved A/B ring remnants from C8 to C9, with addition of a hydroxyl group at C8 (CYP867K1) to generate ginkgolic acid C. This later transformation is likely coupled, however, the mechanism suggested by the authors (Supplemental Data Fig. 3) based on a previously suggested mechanism with different reactants is almost certainly incorrect. Specifically, as that suggested would result in a para-quinone rather than the 8,12-dihydroxylated C ring actually observed (perhaps due to the inaccurate rendering of the C8 carbonyl as a hydroxyl in the originally suggested mechanism?). Given the aromatic nature of the C ring in the ginkgolic acid A substrate, it might worth considering formation of a 8,9-epoxy ring as intermediate instead. As no further transformations were observed with the CYPs from the BGC, co-expression analysis was used to generate a list of 15 CYPs. Notably, this included all 5 from the BGC. Investigation of the remaining 10 led to identification of CYP720B33 as catalyzing formation of a 2,20-olide (yielding ginkgolactone C) and subsequent hydroxylation of C20 to form what is termed ginkgolactone D (please note at this point the minor products yielded by CYP720B31 with levopimaradiene, as shown in Supplemental Fig. 2). It also should be pointed out that that the heterocycle in the latter product could be more accurately termed a lactol and naming this ginkgolactol D might be more informative. Importantly, it seems likely that all the CYP products identified here are intermediates in ginkgolide biosynthesis. However, the suggested further transformations shown in Fig. 5 are arguably a bit misleading as it unclear how the C12 hydroxyl group migrates to C11 in the alternate form of ginkgolactol D. Moreover, a C12 hydroxyl is found in almost all ginkgolides. Thus, it might be more defensible to suggest additional hydroxylation at C11 as the initial transformation with ginkgolactol D, with the indicated heterocyclization otherwise remaining as currently indicated. As noted by the authors, the use of non-seed specific CYP families for ginkgolide biosynthesis is consistent with the 'living fossil' phylogenetic isolation of *G. Biloba*. However, the discussion is arguably overly focused on the importance of BGCs, which is not entirely

consistent with the reported results, specifically the need to also use co-expression analysis, which also identifies all five CYPs from the BGC. This point has been previously made by Wisecarver et al, 2017 (Plant Cell 29:944), which should be cited and the point more clearly made in the discussion here.

Reviewer #2:

Remarks to the Author:

The ginkgolides, a type of diterpene lactones (three lactone groups in ginkgolides), show diverse pharmaceutical activities. However, the ginkgolide biosynthesis pathway remains largely unknown. In this report, Forman et al., functionally characterized five ginkgolide-related P450 enzymes, which are first selected by using BGC-guide and gene-gene coexpression strategies. Given the important bioactivities of ginkgolides, the authors reconstitute the partial pathway of ginkgolides in both microbe and plant chassis. Overall, this a very nice biochemical story, and represent a breakthrough in not only ginkgolide biosynthesis but also plant P450 researches. Below I have provided several points which I think the manuscript can be improved.

1. I can accept that it may not be possible to knockdown expression or knockout genes in *Ginkgo biloba*, but I think it would be appropriate for the authors to acknowledge this limitation of their study in the Discussion section.
2. I am not good at chemical structure elucidation, thus I am wondering how the authors determine the spatial configuration (or chirality) of C2-hydroxyl moiety in compound 8 (Figure S13 and Table S13) and compound 9 (Figure S14 and Table S14) based on the 1D-NMR data. Moreover, the original NMR (¹H and ¹³C) spectrums should be provided as supplemental data, and one additional expert should be invited to evaluate the data.
3. As several novel chemicals had been purified and elucidated in this study, the absolute quantification data could be collected from most experiments, which are also important for future engineering the production of ginkgolides in heterologous hosts.
4. Line 321-322, the observed results here did not completely exclude the possibility that glutathione conjugations to compounds 17 and 23 are enzyme-mediated in tobacco and yeast.
5. In Figure 5, more information should be provide in figure legend to explain the different color used in two chemical structures.
6. I still suggest the author to perform structure modeling (or try with AlphaFold) of the novel P450s discovered in this study to see whether deeper insights of structure-function relationships between key amino acids and the novel activity can be gained from such analysis.

Reviewer #3:

Remarks to the Author:

The manuscript by Forman et al. describes the discovery and characterization of a gene cluster in *Ginkgo biloba* involved in the biosynthesis of ginkgolides. The authors discovered the gene cluster by browsing genomic regions surrounding the putative committed first-step biosynthetic gene GbLPS. They cloned and characterized one diterpene synthase GbLPS and five P450s in the cluster by heterologous expression of combinations of genes in both *N. benthamiana* and yeast. Following extraction, isolation and structure elucidation of enzymatic products from *N. benthamiana* and yeast, the authors successfully elucidated four enzymatic steps from levopimaradiene (2) to ginkgolic acid C, including interesting ring cleavage and oxidative carbon bond shift reactions. The authors further identified a coexpressed P450 enzyme (GbCYP720B31) that could convert ginkgolic acid to ginkgolactone C by furnishing the furan ring. Ginkgolides are structurally complex and intriguing with pharmaceutical and nutraceutical properties. Elucidation of the biosynthetic pathways will provide

insights into how such complex molecules are synthesized in plants and open avenue for metabolic engineering for their production. This work has successfully uncovered a few unique enzymatic steps towards the biosynthesis of ginkgolides with intriguing catalytic activities. Overall, this work is well written and technically sound with proper chemical characterization data. The fact that the six biosynthetic genes described in the manuscript are located in close vicinity to a gene cluster also suggests the evolutionary significance of this pathway. The results presented in this work will be of interest to a broad spectrum of audience in the field of natural product research. I only have a few comments for potential improvement as below:

1. GbLPS identified in this study is a multifunctional diterpene synthase. Is it the only diterpene synthase in the Ginkgo genome that can catalyze the formation compound 2?
2. It would be important to include the GC-MS chromatograms of extracts of tobacco leaves/ yeasts expressing GbLPS+GbCYP7005C1/3 as this could provide clear information on which compounds amongst compounds 1-6 were exactly consumed by the P450s.
3. Would it be possible to propose the structures of glucothione conjugates 20-22 based on the MS2 fragmentation? This would be useful information.

Reviewer #4:

Remarks to the Author:

In the manuscript entitled "A gene cluster in Ginkgo biloba encoding for unique multifunctional cytochrome P450s orchestrates key steps in ginkgolides biosynthesis", the authors used a gene cluster-guided mining approach to reveal the crucial key steps in ginkgolide biosynthesis. The manuscript was well-organized and interesting. However, several concerns need to be addressed.

For all the LC chromatograph, each annotated peak should correspond to an MS spectrum. In addition, each MS spectrum is good to include a chemical structure and present the m/z for quasi-molecular ion and major fragments.

Supplementary Fig.1 should include both the recorded spectra of the metabolites produced by GbLPS in tobacco and yeast and the reference spectra from a database like NIST.

Supplementary Fig. 2 In positive mode, C₂₀H₃₀O₂ will give an m/z 303.2324, C₂₀H₃₂O₃ m/z 321.2430. please explain why the formula C₂₀H₃₀O₂ could be predicted from m/z 301.2166 and 285.2218 under a positive mode. Also please explain why the formulas C₂₀H₃₂O₃ and C₂₀H₃₀O₂ could be predicted from the same m/z 269.2264 under a positive mode.

Supplementary Fig. 3, while the MS2 spectra were presented in Supplementary Fig 4, the MS spectra of compound 7, 11 and 12 should be also included.

L137, Supplementary Fig 11 doesn't show any information about compounds 7 or 8, but only one compound 6. It looks that neither GbCYP7005C1 nor GbCYP7005C3 with GbLPS produced compounds 7 and 8. Please justify how to get a conclusion stated as L135-L137.

L138-L149, Supplementary Fig 2 only shows the results of GbCYP720B33 and GbCYP7005C2, but not three remaining CYPs as indicated. In addition, both produced at least one major peak. Please justify the statement here.

L224, it would be at risk to deduce a molecular structure, even a formula solely based on molecular mass, although by an HRMS. Good to interpret it by integrating quasi-molecular ion, isotope bar, and fragment ions, as well as other supportive information.

Supplementary Table 17, why H-6B has two multiplicity patterns?

Supplementary Table 12-20, pay attention to the font. Most are in regular but some are in bold.

Supplementary Table 20, try to use a NOED or Noesy to distinguish the H-14 from H-15 since H-14 is close to H-9.

L320-322. The results showed that chemical reaction can synthesize the glutathione conjugation of 17 and 23, but it can't exclude the possibility that the glutathione conjugation of 17 and 23 is partly enzyme-driven.

Dear reviewers,

First of all, we would like to sincerely thank you for your positive and constructive comments. Inspired by your remarks we have conducted additional experiments that gave further insight into the enzymatic mechanisms of the novel reported enzymes in this manuscript. We have added the relevant figures and discussion in the submitted revised manuscript. We hope that the new additions have helped us to improve the quality and impact of this manuscript.

Please find below the comments and requests raised by the reviewers in black, and our corresponding responses in blue. We have submitted a PDF file of the manuscript with tracked changes, as well as one without, which we named “clean”. Same stands for the supplemental information.

We trust that you find the introduced changes and our responses to the comments from the reviewers satisfactory. We are looking forward for a positive response and a swift processing,

With best regards,

On behalf of all the authors

REVIEWER COMMENTS and RESPONSES FROM THE AUTHORS

A general notice:

- A. After the agreement with all authors, we have changed the names of the compounds mentioned in this manuscript as follows: “**Ginkgolic acid**” has been replaced by “**Ginkgosinoic acid**”, in accordance with IUPAC nomenclature.
- B. Assoc. Prof. Fenando Geu Flores has been added to the list of co-authors due to his valuable contribution to the conception of the suggested enzymatic mechanisms of the CYP enzymes mentioned in this manuscript.
- C. The numbering of lines in the responses is according to the “clean” text and not the one with tracked changes.

Response to Reviewer #1:

1. First, the almost identical CYP7005C1 and CYP7005C3 both catalyze a mechanistically unusual and fascinating transformation that involves dual scission of the A/B decalin rings, conversion of C2 from the A ring to a carboxylic acid moiety and formation of 1(10) double bond, as well as aromatization of the remaining C ring, forming what the authors term ginkgolic acid A. Two C2-hydroxylated derivatives were observed, 2-hydroxy-levopimaradiene and the further oxidized 2-hydroxy-dehydroabietadiene, with aromatized

C ring. While it seems certain that the first is an intermediate, the status of the second is perhaps less certain than indicated by the authors. Specifically, it might be speculated that cleavage between C9-C10 (B ring), if not also C2-C3 (A ring), could be coupled to C ring aromatization. An experiment is reported indicating dehydroabietadiene is not a substrate for these CYPs. Given that sufficient amounts of the two putative intermediates were isolated for NMR analysis, it would be of interest to determine if these can serve as substrates.

We would like to thank the reviewer for this interesting comment and suggestion. To test whether 2-hydroxy-levopimaradiene (**9**) and 2-hydroxy-dehydroabietadiene (**8**) are substrates for CYP7005C1 and CYP7005C3, we performed yeast feeding experiment using both compounds, **8** and **9**, on yeast strains expressing CYP7005C1 and CYP7005C3. The results can be seen in Extended Data Fig 1. Results clearly show that only 2-hydroxy-levopimaradiene (**9**) can be used as substrate for these CYPs, while 2-hydroxy-dehydroabietadiene (**8**) is not. The findings are discussed in the manuscript (Lines 178-182).

Moreover, we have added a hypothetical enzymatic mechanism of CYP7005C1 and CYP7005C3 (Figure 5b), that we conceived after additional experiments we conducted, as shown in Extended Data Fig. 3. The results are discussed in the text (Lines 414-423).

2. Given that either CYP7005C1 or CYP7005C3 catalyzes this transformation to ginkgolic acid A, it is unclear why all subsequent experiments utilize recombinant co-expression of both (indeed, Supplemental Fig. 3 suggests this decreases yield), at least in tobacco, while no data is presented for yeast

In our subsequent experiments we have included both CYPs, as the two enzymes differ slightly at the C-terminal end, which is a region known to affect the interaction of CYPs with PORs and other CYP enzymes. Thus, we could not rule out a different affinity of these two CYPs towards certain CYPs/PORs, and we deemed it safer to have them both. After all, both enzymes have been maintained in the cluster during Ginkgo evolution, and both enzymes are expressed in ginkgolides producing tissues. We did not include this as a statement in the paper to avoid confusion. Moreover, we do not think the differences observed in Supplemental Fig. 3 are significant, especially considering that tobacco agroinfiltration experiments present a level of variability.

3. as a minor correction the peak in Supplemental Fig. 11 probably should be labeled as '7' rather than '6' and

Thank you for noticing, we have corrected the figure accordingly.

4. the double bonds for miltiradiene in Supplemental Data Fig. 1 are incorrectly positioned
Thank you for the observation, we have corrected the figure accordingly.

5. also please describe construction of the GbLS-MBP fusion in the supplemental methods.
We have added a short cloning description in the supplementary information, page 22, last paragraph.

6. Regardless, it was then found that the two CYP867 family members in the BGC catalyze hydroxylation at C12 (CYP867E38) to form ginkgolic acid B, and, subsequently, the mechanistically interesting 1,2-migration of the cleaved A/B ring remnants from C8 to C9, with addition of a hydroxyl group at C8 (CYP867K1) to generate ginkgolic acid C. This later

transformation is likely coupled, however, the mechanism suggested by the authors (Supplemental Data Fig. 3) based on a previously suggested mechanism with different reactants is almost certainly incorrect. Specifically, as that suggested would result in a para-quinone rather than the 8,12-dihydroxylated C ring actually observed (perhaps due to the inaccurate rendering of the C8 carbonyl as a hydroxyl in the originally suggested mechanism?). Given the aromatic nature of the C ring in the ginkgolic acid A substrate, it might be worth considering formation of a 8,9-epoxy ring as intermediate instead.

We would like to thank the reviewer for the suggestion. To discuss this, we have conducted an additional feeding experiment (Extended Data Fig 3), and we refined our suggestion regarding the hypothetical enzymatic mechanism of GbCYP867K1 as shown in Figure 5c. The results are discussed in the text (Lines 256-268).

7. As no further transformations were observed with the CYPs from the BGC, co-expression analysis was used to generate a list of 15 CYPs. Notably, this included all 5 from the BGC. Investigation of the remaining 10 led to identification of CYP720B31 as catalyzing formation of a 2,20-olide (yielding ginkgolactone C) and subsequent hydroxylation of C20 to form what is termed ginkgolactone D (please note at this point the minor products yielded by CYP720B31 with levopimaradiene, as shown in Supplemental Fig. 2). It also should be pointed out that the heterocycle in the latter product could be more accurately termed a lactol and naming this ginkgolactol D might be more informative.

We would like to keep the name as is, because chemically, this is an α,β -unsaturated γ -lactone with an additional hydroxyl group in the γ -position – formed by the nucleophilic attack of the carboxylic acid OH group (in the open form) at the carbonyl group of an aldehyde at C-20. However, since the carboxylic acid (as well as the lactone) is higher oxygenated than both the C-20 of the lactol, chemical rules tell that the lactone takes priority in terms of naming the compound.

8. Importantly, it seems likely that all the CYP products identified here are intermediates in ginkgolide biosynthesis. However, the suggested further transformations shown in Fig. 5 are arguably a bit misleading as it is unclear how the C12 hydroxyl group migrates to C11 in the alternate form of ginkgolactol D. Moreover, a C12 hydroxyl is found in almost all ginkgolides. Thus, it might be more defensible to suggest additional hydroxylation at C11 as the initial transformation with ginkgolactol D, with the indicated heterocyclization otherwise remaining as currently indicated.

We apologize for the confusion. We hope that the updated Fig. 5a will be more informative and can make our suggestion more clear. We have numbered the carbon atoms in ginkgolactone D to show that we are simply rotating the molecule to showcase how the hydroxylations at C12, C8 and C11 can potentially fit with the structure of ginkgolides, according to our hypothesis.

9. As noted by the authors, the use of non-seed specific CYP families for ginkgolide biosynthesis is consistent with the 'living fossil' phylogenetic isolation of *G. biloba*. However, the discussion is arguably overly focused on the importance of BGCs, which is not entirely consistent with the reported results, specifically the need to also use co-expression analysis, which also identifies all five CYPs from the BGC. This point has been previously made by Wisecaver et al, 2017 (Plant Cell 29:944), which should be cited and the point more clearly made in the discussion here.

The reviewer is right in his comment, and we apologize. Our excessive focus on the gene clusters is probably the result of our enthusiasm when we realized the presence of BGCs in

Ginkgo biloba, more over that this cluster led us to the identification of the first CYPs in ginkgolides biosynthesis. Nevertheless, it is true that co-expression analysis has been proven a powerful tool as well, as all CYPs in the cluster were also found co-expressed with GbLPS, and it helped us identify novel CYP enzymes not included in the cluster. Therefore, we have done certain changes in the manuscript to reflect this. Changes are highlighted by “tracked-changes”.

Response to Reviewer #2:

1. I can accept that it may not be possible to knockdown expression or knockout genes in *Ginkgo biloba*, but I think it would be appropriate for the authors to acknowledge this limitation of their study in the Discussion section.

Thank you for the suggestion, we have added a sentence about this at the discussion, Lines 470-473.

2. I am not good at chemical structure elucidation, thus I am wondering how the authors determine the spatial configuration (or chirality) of C2-hydroxyl moiety in compound 8 (Figure S13 and Table S13) and compound 9 (Figure S14 and Table S14) based on the 1D-NMR data. Moreover, the original NMR (1H and 13C) spectrums should be provided as supplemental data, and one additional expert should be invited to evaluate the data.

We have determined the spatial configuration of C2-hydroxyl moiety in compound 8 (Figure S13 and Table S13) and compound 9 (Figure S14 and Table S14) according to the following: The fused cyclohexane A-ring is in a chair conformation, with H-5 and CH₃-20 in trans diaxial configurations. The α -orientation of the OH-group is determined based on the coupling pattern and coupling constant of H-2 β . Thus, H-2 β (axially oriented) display a triplet of triplets, with axial, axial couplings (11.5 Hz) to H-1 α (axially oriented) and H-3 α (axially oriented) and with axial, equatorial couplings (4.1 Hz) to H-1 β (equatorially oriented) and H-3 β (equatorially oriented). This proves that H-2 is in β -orientation (axially up) – and consequently the OH group α -oriented. This is further confirmed by the axial, axial couplings observed between H-2 β and H-3 α . Regarding NMR spectra, we did not include them, as we believe they are not exactly relevant for the reader, although they are available. In case the editor and the reviewer find it necessary, we could include them.

3. As several novel chemicals had been purified and elucidated in this study, the absolute quantification data could be collected from most experiments, which are also important for future engineering the production of ginkgolides in heterologous hosts.

According to the suggestion of the reviewer, we have quantified the yeast heterologous production of ginkgosinoic acid B (**11**), as it was the only compound available in sufficient amounts, for that purpose. We did not proceed to the purification of other compounds exclusively for quantification, as we believe that the produced amounts are indicative, and we have not performed experiments for production optimization and metabolic engineering of the yeast strains. We hope that this can answer the question of the reviewer and give an indication of the level of the titers of the produced terpenoids in this article. The results are reported in Line 233.

4. Line 321-322, the observed results here did not completely exclude the possibility that glutathione conjugations to compounds 17 and 23 are enzyme-mediated in tobacco and yeast.

Thank you for the comment, it is true that glutathione conjugation could be facilitated by enzymes when studies are in vivo, therefore, we have rephrased the sentence to show this possibility as well, in the main text, Lines 345-347.

5. In Figure 5, more information should be provided in figure legend to explain the different color used in two chemical structures

Thank you for these suggestions – we have added a small sentence for clarification.

6. I still suggest the author to perform structure modeling (or try with AlphaFold) of the novel P450s discovered in this study to see whether deeper insights of structure-function relationships between key amino acids and the novel activity can be gained from such analysis.

This is a really interesting suggestion, however, unfortunately did not yield any meaningful results we could use to substantiate our findings. We hope to be able to use this methodology in the future, in combination with mutagenesis and substrate docking studies to establish a better understanding of the interesting activities of the CYP7005C1 and C3, as well as the CYP867K1 and to uncover the mechanism of their unusual reactions.

Response to Reviewer #3:

1. GbLPS identified in this study is a multifunctional diterpene synthase. Is it the only diterpene synthase in the Ginkgo genome that can catalyze the formation compound 2?

The GbLPS mentioned in this manuscript is the only diterpene synthase found in Ginkgo transcriptomes or genome. There is a recent paper describing another putative levopimaradiene synthase from Ginkgo (<https://doi.org/10.1080/15592324.2021.1885906>), however, no biochemical characterization of this enzyme is described. And although we identified the locus encoding this putative enzyme in the genome of Ginkgo, it was not possible to amplify, and clone thereafter, the corresponding cDNA. Nevertheless, studying the amino acid sequence of the deduced protein, it resembles more to a sesquiterpene synthase than to a diterpene, according to the identified signature-motifs.

2. It would be important to include the GC-MS chromatograms of extracts of tobacco leaves/ yeasts expressing GbLPS+GbCYP7005C1/3 as this could provide clear information on which compounds amongst compounds 1-6 were exactly consumed by the P450s.

Interestingly, all products of GbLPS are lowered when GbCYP7005C1/C3 are co-expressed, thus making it impossible to determine which substrate has been used. We assume this phenomenon is due to the reaction mechanism and the product of GbLPS, which possibly, like other bifunctional diterpene synthases from gymnosperms, produces an unstable hydroxylated product that can form 2-6 when submitted to GC-MS analysis, due to dehydration. This has been shown nicely from Keeling et al., 2011 (DOI:<https://doi.org/10.1074/jbc.M111.245951>). We speculate that formation of 2 might likely be in equilibrium with products 3-6. If 2 is taken up by GbCYP7005C1/C3, this might overall decrease the availability of all compounds, 2-6, masking what the “real” substrate is. We have left out this data and information for now as it

did not add any additional information to the manuscript, in our opinion, and might create a confusion for the reader, but we could include it if the reviewer and editor find it relevant. To identify the real substrate of GbCYP7005C1/C3 we have performed different type of experiments that can be seen in “Extended Data Fig. 1”.

3. Would it be possible to propose the structures of glutathione conjugates 20-22 based on the MS2 fragmentation? This would be useful information.

Thank you for this suggestion. After evaluating the fragmentation patterns of the MS2 data of these compounds, we show the suggested compounds at Supplementary Fig.10c.

Response to Reviewer #4:

1. For all the LC chromatograph, each annotated peak should correspond to an MS spectrum. In addition, each MS spectrum is good to include a chemical structure and present the m/z for quasi-molecular ion and major fragments.

Thank you for the suggestion. We have included the MS spectra of the main molecules discussed throughout this article and their structures in Supplementary Figure 21.

2. Supplementary Fig.1 should include both the recorded spectra of the metabolites produced by GbLPS in tobacco and yeast and the reference spectra from a database like NIST.

We have included all spectra recorded from yeast and tobacco cells expressing GbLPS, and NIST library data hits when found. For the two compounds without NIST reference (Compounds 6 and 10), we have referred to articles that have reported spectra of these molecules.

3. Supplementary Fig. 2 In positive mode, C₂₀H₃₀O₂ will give an m/z 303.2324, C₂₀H₃₂O₃ m/z 321.2430. please explain why the formula C₂₀H₃₀O₂ could be predicted from m/z 301.2166 and 285.2218 under a positive mode. Also please explain why the formulas C₂₀H₃₂O₃ and C₂₀H₃₀O₂ could be predicted from the same m/z 269.2264 under a positive mode.

Thank you for noticing. We had indeed made a mistake while analyzing the data represented in Supplementary Fig. 2 – we have now included the MS spectra of the molecules produced by GbCYP720B33 (compounds **i** and **ii**) and corrected their predicted formulas to match the correct quasi-molecular ion. For GbCYP7005C2, after reanalyzing the data, we realized that the enzyme is producing small amounts of molecule **8** (2-hydroxydehydroabietadiene). This has been added in the main text of the manuscript as well. For molecule **8**, the quasi-molecular ion is easily lost, therefore we used instead m/z 269.2268 as a scan ion.

4. Supplementary Fig. 3, while the MS2 spectra were presented in Supplementary Fig 4, the MS spectra of compound 7, 11 and 12 should be also included.

The spectra of the above compounds have been added to Supp. Fig 21.

5. L137, Supplementary Fig 11 doesn't show any information about compounds 7 or 8, but only one compound 6. It looks that neither GbCYP7005C1 nor GbCYP7005C3 with GbLPS produced compounds 7 and 8. Please justify how to get a conclusion stated as L135-L137.

Sorry for the mistake, compound annotated as **6** at Supp. Figure 11 should be **7**, therefore it is the main product of GbCYP7005C1 or GbCYP7005C3. At the original figure, we had not included the ion corresponding to compound **8** (m/z 269.2268), and this why **8** was not visible. We have

updated the figure where compound **8** is visible, as a product of GbCYP7005C1 and GbCYP7005C3. We have updated the figure legend as well.

6. L138-L149, Supplementary Fig 2 only shows the results of GbCYP720B33 and GbCYP7005C2, but not three remaining CYPs as indicated. In addition, both produced at least one major peak. Please justify the statement here.

Sorry for the confusion. We believe that this misinterpretation was due to wrong phrasing. We have edited the sentence to make it clear to the reader (Lines 141 -144). Supp. Fig. 2 does not show all 3 remaining CYPs of the cluster, but only the one that gave a detectable product (CYP7005C2), when co-expressed with GbLPS. GbCYP720B33 is not part of the cluster, but it is included in the figure as it produced two minor peaks when co-expressed with GbLPS, and it is included in the GbLPS co-expression module. Therefore, we considered the information relevant for future experiments. We have also added an extra sentence at the figure legend.

7. L224, it would be at risk to deduce a molecular structure, even a formula solely based on molecular mass, although by an HRMS. Good to interpret it by integrating quasi-molecular ion, isotope bar, and fragment ions, as well as other supportive information.

The molecular formula identification of all new products (Table 1) was performed using Bruker's built-in software, taking the mass error (in ppm) and the true isotope pattern into consideration – and the same for fragment ions where available. The mass errors are all below 5 ppm, and for most of the products in Table 1, well below 1 ppm. Therefore, we believe, this leaves no doubt about the stated molecular formula. The suggestion of 14-16 being glutathione conjugates comes from the fact that when compound 17 is incubated in vitro with glutathione, it gives exactly the same products as we see in Fig.3b. (Supplementary Fig 9a).

8. Supplementary Table 17, why H-6B has two multiplicity patterns?

Thanks for noticing. The '(1H, m)' has been deleted.

9. Supplementary Table 12-20, pay attention to the font. Most are in regular but some are in bold.

Thanks for noticing. The bold formatting has been removed.

10. Supplementary Table 20, try to use a NOED or Noesy to distinguish the H-14 from H-15 since H-14 is close to H-9.

There is free rotation around the C1-C13 single bond, and thus the figure merely illustrates one out of many rotamers. Thus, NOESY or alike experiments will not be able to discriminate between the two diastereotopic methyl groups H-14 and H-15.

11. L320-322. The results showed that chemical reaction can synthesize the glutathione conjugation of 17 and 23, but it can't exclude the possibility that the glutathione conjugation of 17 and 23 is partly enzyme-driven.

Thank you for the comment, it is true that glutathione conjugation could be facilitate by enzymes when studies are in vivo, therefore, we have rephrased the sentence to show this possibility as well, in the main text, Lines 345-347.

Reviewers' Comments:

Reviewer #1:

Remarks to the Author:

This manuscript has been adequately revised and now has a firmer biochemical foundation. In discussion of CYP mediated rearrangement (Line 425), while understanding that the authors desire to reference their own work, it might be best to add the arguably more relevant (diterpenoid) example from gibberellin biosynthesis. Specifically, the ring contraction reaction catalyzed by CYP88. There is an error in Fig 2g, where the 12-hydroxyl needs to be removed from ginkgosinoic acid A. Also, it would be useful to include the NMR spectra as requested by another reviewer. Particularly given this, it would helpful to provide a table of contents for the supplementary information document.

Reviewer #2:

Remarks to the Author:

I appreciate the authors' work for manuscript improvement and I have no further suggestions.

Reviewer #3:

Remarks to the Author:

This reviewer's concerns have been fully resolved.

Reviewer #4:

Remarks to the Author:

The authors addressed most of my questions.

Dear reviewers,

Thank you again for your positive and quick replies.

We have submitted a manuscript with your suggested changes that can be seen by “tracked changes”.

With best regards,

On behalf of all the authors

REVIEWER COMMENTS and RESPONSES FROM THE AUTHORS

Response to Reviewer #1:

- This manuscript has been adequately revised and now has a firmer biochemical foundation. In discussion of CYP mediated rearrangement (Line 425), while understanding that the authors desire to reference their own work, it might be best to add the arguably more relevant (diterpenoid) example from gibberellin biosynthesis. Specifically, the ring contraction reaction catalyzed by CYP88.

Recognizing that the comment of the reviewer is appropriate and the example they are suggesting is more suitable in this case than the one we have used, we have updated the text of the manuscript and included as reference the suggested CYP88 enzyme. Changes are tracked.

- There is an error in Fig 2g, where the 12-hydroxyl needs to be removed from ginkgosinoic acid A.

Sorry for the mistake, we have updated the Figure 2g.

- Also, it would be useful to include the NMR spectra as requested by another reviewer. Particularly given this, it would be helpful to provide a table of contents for the supplementary information document.

We have included the NMR spectra and have made a “table of contents” for the supplemental Information.

Response to Reviewer #2:

I appreciate the authors’ work for manuscript improvement and I have no further suggestions

Response to Reviewer #3:

This reviewer's concerns have been fully resolved

Response to Reviewer #4:

The authors addressed most of my questions.